# Achieving $\tilde{O}(1/\varepsilon)$ Sample Complexity for Constrained Markov Decision Process

**Jiashuo Jiang**
Department of Industrial Engineering & Decision Analytics
Hong Kong University of Science and Technology
Hong Kong, China
jsjiang@ust.hk

**Yinyu Ye**
Department of Management Science & Engineering
Institue of Computational Mathematics and Engineering
Stanford University
California, US
yyye@stanford.edu

## Abstract

We consider the reinforcement learning problem for the constrained Markov decision process (CMDP), which plays a central role in satisfying safety or resource constraints in sequential learning and decision-making. In this problem, we are given finite resources and a MDP with unknown transition probabilities. At each stage, we take an action, collecting a reward and consuming some resources, all assumed to be unknown and need to be learned over time. In this work, we take the first step towards deriving optimal problem-dependent guarantees for the CMDP problems. We derive a logarithmic regret bound, which translates into a $O(\frac{1}{\Delta \cdot \varepsilon} \cdot \log^2(1/\varepsilon))$ sample complexity bound, with $\Delta$ being a problem-dependent parameter, yet independent of $\varepsilon$. Our sample complexity bound improves upon the state-of-art $O(1/\varepsilon^2)$ sample complexity for CMDP problems established in the previous literature, in terms of the dependency on $\varepsilon$. To achieve this advance, we develop a new framework for analyzing CMDP problems. To be specific, our algorithm operates in the primal space and we resolve the primal LP for the CMDP problem at each period in an online manner, with *adaptive* remaining resource capacities. The key elements of our algorithm are: i) a characterization of the instance hardness via LP basis, ii) an eliminating procedure that identifies one optimal basis of the primal LP, and; iii) a resolving procedure that is adaptive to the remaining resources and sticks to the characterized optimal basis.

## 1 Introduction

Reinforcement learning (RL) is pivotal in the realm of dynamic decision-making under uncertainty, where the objective is to maximize total reward through ongoing interaction with and learning from an enigmatic environment. Markov Decision Processes (MDPs) are a prevalent framework for encapsulating environmental dynamics. MDPs have been instrumental in various domains, such as video gaming [52], robotics [41], recommender systems [57], inventory control [11], and beyond. Yet, they fall short in accommodating additional constraints that may influence the formulation of the optimal policy and the decision-maker's engagement with the uncertain environment. Often, in MDP applications, there are stringent constraints on utilities or costs, emanating from areas like safe

38th Conference on Neural Information Processing Systems (NeurIPS 2024).

autonomous driving [22], robotics [56], revenue management [33] and financial management [62]. These constraints might also symbolize limitations on resources within resource allocation applications. Constrained MDPs (CMDPs), as introduced in [4], enhance MDPs to factor in constraints affecting long-term policy results. In CMDPs, the decision-maker aims to optimize cumulative rewards while adhering to these constraints. Our paper focuses on CMDPs, and we aim to develop efficient algorithmic solutions.

The significance of RL in CMDP contexts has garnered substantial attention in recent years. A variety of methods for tackling CMDPs have been developed, including the primal-dual technique [21, 49, 12, 69, 18, 45, 25, 26], which leverages the Lagrangian dual of CMDPs and implements an online learning strategy for the iterative update of dual variables. Other methods encompass constrained optimization [1, 61], the Lyapunov technique [14], among others. Previous research has established *minimax* bounds for CMDPs, representing the optimal regret that can be achieved for the most challenging problem within a specific class of problems. Nonetheless, these minimax regret bounds and worst-case scenarios can be overly conservative, leading to a disconnect between theoretical guarantees and practical performance for specific problem instances. A more tailored approach is warranted—one that ensures great performance on every single problem instance and furnishes problem-dependent guarantees. Our research takes the first step towards deriving optimal problem-dependent guarantees for CMDP problems.

## 1.1 Preliminaries

We consider a CMDP problem with a finite set of states $\mathcal{S} = \{1, 2, \ldots, |\mathcal{S}|\}$ and a finite set of actions $\mathcal{A} = \{1, 2, \ldots, |\mathcal{A}|\}$. We denote by $\gamma \in (0, 1)$ a discount factor. We also denote by $P : \mathcal{S} \times \mathcal{A} \to \mathcal{D}(\mathcal{S})$ the probability transition kernel of the CMDP, where $\mathcal{D}(\mathcal{S})$ denotes a probability measure over the state space $\mathcal{S}$. Then, $P(s'|s, a)$ denotes the probability of transiting from state $s$ to state $s'$ when the action $a$ is executed. The initial distribution over the states of the CMDP is denoted by $\mu_1$.

There is a *stochaastic* reward function $r : \mathcal{S} \times \mathcal{A} \to \mathcal{D}[0, 1]$ and $K$ *stochastic* cost functions $c_k : \mathcal{S} \times \mathcal{A} \to \mathcal{D}[0, 1]$ for each $k \in [K]$. We also denote by $\hat{r}(s, a) = \mathbb{E}[r(s, a)]$ for each $(s, a)$ and $\hat{c}(s, a) = \mathbb{E}[c_k(s, a)]$ for each $(s, a)$. For any Markovian policy $\pi$, where the action of $\pi$ depends only on the current state and the action of $\pi$ is allowed to be randomized, we denote by $V_r(\pi, \mu_1)$ the infinite horizon discounted reward of the policy $\pi$, with the formulation of $V_r(\pi, \mu_1)$ given below:

$$V_r(\pi, \mu_0) = \mathbb{E}\left[\sum_{t=0}^{\infty} \gamma^t \cdot r(s_t, a_t) \mid \mu_1\right],$$  (1)

where $(s_t, a_t)$ is generated according to the policy $\pi$ and the transition kernel $P$ with the initial state distribution $\mu_1$. For each $k \in [K]$, the infinite horizon discounted cost of the policy $\pi$ is denoted by $V_k(\pi, \mu_1)$, and the following constraint needs to be satisfied by the policy $\pi$,

$$V_k(\pi, \mu_1) = \mathbb{E}\left[\sum_{t=0}^{\infty} \gamma^t \cdot c_k(s_t, a_t) \mid \mu_1\right] \leq \alpha_k, \ \forall k \in [K].$$  (2)

To solve the CMDP problem, we aim to find an optimal Markovian policy, denoted by $\pi^*$, that maximizes the reward in (1) while satisfying the cost constraint (2) for each $k \in [K]$, with $\alpha_k \in \left[0, \frac{1}{1-\gamma}\right]$ being a pre-specified value for each $k \in [K]$. Importantly, we assume that the reward function $r$, the cost functions $\{c_k\}_{k=1}^{K}$, and the transition kernel $P$, are all **unknown** to the decision maker. Our goal is to obtain a policy $\pi$ that approximates the optimal policy $\pi^*$ with as few samples as possible. We now describe the sampling procedure and present the performance measure of our policy. We assume the existence of a stylized *generative model* $\mathcal{M}$, as studied in [39, 38, 44]. The model $\mathcal{M}$ satisfies the following condition.

**Assumption 1.1** *For each state and action pair $(s, a)$, we can query the model $\mathcal{M}$ to obtain an observation of $r(s, a)$, $c_k(s, a)$ for each $k \in [K]$, and the new state $s' \in \mathcal{S}$, where the transition from $s$ to $s'$ follows the probability kernel $P(s'|s, a)$ independently.*

Note that in reinforcement learning for CMDP problems, querying the generative model $\mathcal{M}$ can be costly. Therefore, it is desirable to query the model $\mathcal{M}$ as less as possible, while guaranteeing the

near optimality of the approximate policy. To this end, we use *sample complexity* as the measure of the performance of our policy. For any $\varepsilon$, we aim to find an $\varepsilon$-accurate policy $\pi$ such that

$$V_r(\pi^*, \mu_1) - V_r(\pi, \mu_1) \leq \varepsilon \ \text{ and } \ V_k(\pi, \mu_1) - \alpha_k \leq \varepsilon, \ \forall k \in [K], \tag{3}$$

with as few samples as possible.

## 1.2 Our Main Results and Contributions

The main result of our research is the introduction of a novel algorithm that promises a $O(\frac{1}{\Delta \cdot \varepsilon} \cdot \log^2(1/\varepsilon))$ sample complexity bound, where $\Delta$ is a positive constant that characterizes the gap between the optimal policy and the sub-optimal ones. Note that the state-of-the-art sample complexity bound under the worst-case scenarios is $O(1/\varepsilon^2)$, which has been established in a series of work [68, 37, 21]. Though the $O(1/\varepsilon)$ or better iteration complexity has been achieved in [45, 50, 74, 26], it comes with a sample complexity bound no better than $O(1/\varepsilon^2)$. Our algorithm enjoys a sample complexity bound that has a better dependency in terms of $\varepsilon$. To achieve this improved result, we develop several new elements listed below.

**Contribution 1**: we develop new characterizations of the problem instance hardness for CMDP problems. Note that a key component for achieving instance-dependent bounds is to characterize the "hardness" of the underlying problem instance. That is, we need to identify a positive gap to separate the optimal policy from the sub-optimal policies for a particular problem instance. The importance of identifying such a gap has been demonstrated in instance-optimal learning for multi-arm-bandit problems (e.g. [42]) and reinforcement learning problems (e.g. [6]), among others. For CMDP, identifying such a gap is non-trivial because the optimal policies for CMDP are randomized policies [4]. Then, the policies can be represented by distributions over the action set and the sub-optimal policies can be arbitrarily close to the optimal policy as long as the corresponding sub-optimal distributions converge to the optimal one. To tackle this problem, we show that the feasible region for the policies can be represented as a polytope and we only need to focus on the corner points of this polytope to find an optimal policy. Therefore, the hardness can simply be characterized as the distance between the optimal corner point and the sub-optimal corner point, as illustrated in detail in Section 2.1. This is the first characterization of problem instance hardness for CMDP problems.

**Contribution 2**: we devise a new algorithmic framework to analyze CMDP problems, inspired by the online packing/linear programming (LP) literature [3, 40, 47, 46]. Specifically, we utilize a linear programming reformulation of the CMDP problem, where policies are delineated via *occupancy measures* [4]. The optimal policy emerges from the LP's solution; however, the indeterminate model parameters mean the LP cannot be solved directly but must be approached online as we obtain more samples from the generative model. Each generative model query leads to solving an empirical LP with accrued samples, and our final policy is derived from averaging these solutions—an approach akin to the methodology in online LP. A critical feature of our algorithm is the adaptiveness of the LP constraints' right-hand side to the input samples, a technique demonstrated to achieve logarithmic regret in online LP literature, which we now apply to CMDP problems.

**Contribution 3**: we extend our contributions to the online LP literature. Note that after adopting the LP reformulation, the corner points of the feasible region for policies can be represented by the basis of the LP. Separating the optimal policy from the others is equivalent to identifying one optimal basis of the LP. We utilize an approach that lexicographically restricts one variable to zero and tests whether the LP value has changed. We show that this approach systematically pinpoints a particular optimal LP basis with a high probability. Then, we develop a resolving procedure that capitalizes on the structure of the identified optimal basis, which involves only the non-zero basic variables and the active constraints. This is a new approach of deriving problem-dependent bound for online LP.

**Other related literature.** Problem-dependent guarantees have been considered extensively in the RL literature, where a series of work [72, 59, 54, 23, 17, 66, 65, 71, 20] establishes the $\log(N)$ regret bound or the $O(\kappa \cdot \varepsilon^{-1})$ sample complexity, with $\kappa$ being a problem-dependent constant. Our approach can also be directly applied to the RL problems. Moreover, our approach can handle long-term constraints and can deal with multi-objective (safe) RL problems.

our work presents a new algorithm for safe RL problems. We adopt an occupancy measure representation of the optimal and obtain an LP to work with, which is similar to the previous work. However, our algorithm resolves an LP and operates in the primal space, which is fundamentally different from the previous work that adopts a primal-dual update (e.g. [60], [19], [73], [8], [53]). There is also

work developing primal-based algorithms, for example ([51], [13], [15], [16], [70]). Our algorithm is completely different from the previous work and we obtain new results. The result is that we are able to obtain an instance-dependent $\tilde{O}(1/\epsilon)$ sample complexity the first time in the literature, which improves upon the $O(1/\epsilon^2)$ worst-case sample complexity established in the previous work. Though the constrained optimization approach and the Lyapunov approach have also been developed for CMDP problems, they do not enjoy a theoretical guarantee. In comparison to the literature, we develop a new primal-based algorithm and achieve the first instance-dependent sample complexity for CMDP problems.

Our new primal-based algorithm is motivated from the literature of online linear programming [3, 40, 47] and bandits with knapsack problems (e.g. [46], [48]). In these problems, the optimal policy can be written as an LP and we need to develop an online policy to solve this LP sequentially. Note that a prevalent strategy is to resolve the LP adaptive to the remaining resources, which has been developed in a long line of research on various applications, for example [31], [32], [5], [36], [64], [35] and [10]. We make the innovation of resolving the LP while sticking to the identified optimal basis, which distinguishes our algorithm from the previous ones in online LP. Note that the online LP techniques has also been extended to handle non-stationarity, for example in [9], [34]. It is an interesting future topic to explore whether our algorithm can be extended to non-stationary environment.

## 2 LP Reformulation

The infinite horizon discounted setting described in Section 1.1 admits a linear programming reformulation. To be specific, due to the existence of the constraints, the optimal policy of a CMDP can be randomized policies (e.g. [4]), where it is optimal to take a stochastic action given the current state. Therefore, it is convenient to represent a policy through the *occupation measure*, which gives us the desired linear programming reformulations of the CMDP problems.

For the infinite horizon discounted problem, the occupancy measure is defined as $q_\pi(s, a)$ for any state $s$, action $a$, and policy $\pi$. Note that $q_\pi(s, a)$ represents the total expected discounted time spent on the state-action pair $(s, a)$, under policy $\pi$, multiplied by $1 - \gamma$. Then, following [4], the optimal policy (and the optimal occupancy measure) can be obtained from the following linear programming.

$$V^{\text{Infi}} = \max \sum_{s \in \mathcal{S}} \sum_{a \in \mathcal{A}} \hat{r}(s, a) \cdot q(s, a) \tag{4a}$$

$$\text{s.t.} \sum_{s \in \mathcal{S}} \sum_{a \in \mathcal{A}} \hat{c}_k(s, a) \cdot q(s, a) \leq \alpha_k \qquad \forall k \in [K] \tag{4b}$$

$$\sum_{s' \in \mathcal{S}} \sum_{a \in \mathcal{A}} q(s', a) \cdot (\delta_{s,s'} - \gamma \cdot P(s|s', a)) = (1 - \gamma) \cdot \mu_1(s) \qquad \forall s \in \mathcal{S} \tag{4c}$$

$$q(s, a) \geq 0 \qquad \forall s \in \mathcal{S}, a \in \mathcal{A}, \tag{4d}$$

where $\delta_{s,s'} = \mathbb{1}_{s=s'}$, and $\mu_1(s)$ denotes the probability for the first state to be realized as $s \in \mathcal{S}$ following the initial distribution $\mu_1$. Note that for an optimal solution $\{q^*(s, a)\}_{\forall s \in \mathcal{S}, \forall a \in \mathcal{A}}$ to (4), the corresponding optimal policy $\pi^*$ will be

$$P(\pi^*(s) = a) = \begin{cases} \dfrac{q^*(s, a)}{\sum_{a' \in \mathcal{A}} q^*(s, a')}, & \text{if } \sum_{a' \in \mathcal{A}} q^*(s, a') > 0 \\ 1/|\mathcal{A}|, & \text{if } \sum_{a' \in \mathcal{A}} q^*(s, a') = 0, \end{cases} \tag{5}$$

where $\pi^*(s)$ denotes the probability for the policy $\pi^*$ to take the action $a \in \mathcal{A}$ given the state $s \in \mathcal{S}$. In fact, when $\sum_{a' \in \mathcal{A}} q^*(s, a') = 0$, we can take an arbitrary action. In what follows, we rely on the linear programming formulation (4) to derive our results.

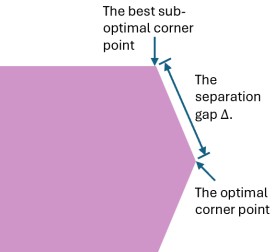

The best sub-optimal corner point

The separation gap Δ.

The optimal corner point

Figure 1: A graph illustration of the hardness characterization via LP basis, where the shaded area denotes the feasible region for the policies.

## 2.1 Characterization of Instance Hardness

We now rewrite the LP (4) into the following standard formulation to proceed with our illustration.

$$
\begin{aligned}
V = \max \quad & \hat{\boldsymbol{r}}^\top \boldsymbol{q} \\
\text{s.t.} \quad & C\boldsymbol{q} \le \boldsymbol{\alpha} \\
& B\boldsymbol{q} = \boldsymbol{\mu} \\
& \boldsymbol{q} \ge 0.
\end{aligned}
\tag{6}
$$

Note that one crucial step for achieving problem-dependent bounds is to characterize the hardness of the underlying problem instance and define a gap that separates the optimal policies from the others. For multi-arm-bandit problem, the characterization of the hardness can be the gap between the optimal arm and the best sub-optimal arm (e.g. [42]). For reinforcement learning problem, the characterization of the hardness can also be the gap between the optimal policy and the best sub-optimal policy (e.g. [6]). As long as this separation gap is a positive constant, denoted by $\Delta$, separating the optimal policy from the others with a probability at least $1 - \epsilon$ would require samples at most $\frac{1}{\Delta} \cdot \log(1/\epsilon)$, which finally implies the instance-optimal sample complexity bound.

For CMDP problem, characterizing the hardness of the problem instance can be hard. Based on LP (6), we know that a feasible policy corresponds to a feasible solution and the sub-optimal solution can be arbitrarily close to the optimal solution since the feasible set is "continuous". Therefore, there is no direct way to identify a positive gap between the optimal policies and the sub-optimal ones. However, from standard LP theory, we know that one **corner point** of the feasible region must be one optimal solution. Therefore, we can simply focus on the corner points when solving LP (6) and we define the gap as *the distance between the optimal corner point and the sub-optimal corner point*, as illustrated in Figure 1. As we will show later, the LP reformulation (6) and such a characterization of hardness via corner points will inspire our entire approach.

Since the problem hardness is characterized via corner points, it is essential to provide further characterization of the corner points. Note that in LP theory, the corner point is called *basic solution* and can be represented by *LP basis*, which involves the set of basic variables that are allowed to be non-zero, and the set of active constraints that are binding under the corresponding basic solution. Our next lemma follows from standard LP theory, where the proof is provided in the appendix for completeness.

**Lemma 2.1** *Denote by $b$ the number of rows in the matrix $B$. Then, there exists a subset $\mathcal{J}^* \subset [K]$ with $K' = |\mathcal{J}^*|$ and an optimal solution $\boldsymbol{q}^*$ to LP (6) such that there are $b + K'$ variables in $\boldsymbol{q}^*$ are non-zero. Moreover, we denote by $\mathcal{I}^*$ the index set of the non-zero element in $\boldsymbol{q}^*$. Then, the optimal solution $\boldsymbol{q}^*$ can be uniquely determined as the solution to the linear system*

$$
C(\mathcal{J}^*, \mathcal{I}^*)\boldsymbol{q}_{\mathcal{I}^*} = \boldsymbol{\alpha}_{\mathcal{J}^*}, \tag{7a}
$$

$$
B(:, \mathcal{I}^*)\boldsymbol{q}_{\mathcal{I}^*} = \boldsymbol{\mu}, \tag{7b}
$$

$$
\boldsymbol{q}_{\mathcal{I}^{*c}} = 0. \tag{7c}
$$

*with $\mathcal{I}^{*c}$ being the complementary set of the index set $\mathcal{I}^*$.*

**Remark**. Note that the minimax lower bound $\Omega(1/\epsilon^2)$ has been established in previous work [7, 63]. However, this does not contradict with our $\tilde{O}(1/\epsilon)$ sample complexity after we introduce the instance

hardness measure $\Delta$. To be specific, for a problem instance $I$, we can denote by $S(I, \epsilon)$ the number of samples needed to construct an $\epsilon$-optimal policy. Then the worst-case lower bound implies that $\max_I S(I, \epsilon) = \Theta(1/\epsilon^2)$. However, if we do not consider the worst-case guarantee, i.e., if we do not maximize over the problem instance $I$, then we can characterize an instance-dependent constant $\Delta(I)$ (independent of $\epsilon$) such that $S(I, \epsilon) = \Delta(I)/\epsilon \cdot \text{polylog}(1/\epsilon)$. When the problem instance is favorable such that the constant $\Delta(I)$ is smaller than $1/\epsilon$, our bound strictly improves upon the worst-case bound.

**Overview of our approach**: in the first part, we aim to identify the optimal basis $\mathcal{I}$ and $\mathcal{J}$. In this way, we identify the optimal corner point to look at. The detailed procedure is described in Section 3. In the second part, we learn the optimal solution given the optimal basis we have identified, which finally gives us a near-optimal policy with the desired sample complexity bound. The detailed procedure is described in Section 4.

## 3 Construct Estimates and Identify Optimal Basis

We describe how to construct estimates for LP (6). To this end, for a round $N_0$, we denote by $\mathcal{F}_{N_0}$ the filtration of all the information collected up to round $N_0$. Then, we denote by $\bar{C}_{N_0}$ (resp. $\bar{B}_{N_0}$) an estimate of the matrix $C$ (resp. $B$), constructed using the information in the set $\mathcal{F}_{N_0}$. We also denote by $\bar{\boldsymbol{r}}_{N_0}$ and estimate of $\hat{\boldsymbol{r}}$ constructed from the information in the set $\mathcal{F}_{N_0}$. To be specific, we define

$$\bar{r}_{N_0}(s, a) = \frac{\sum_{n=1}^{N_0} r^n(s, a)}{N_0}, \bar{c}_{k, N_0}(s, a) = \frac{\sum_{n=1}^{N_0} c_k^n(s, a)}{N_0}, \text{ and } \bar{P}_{N_0}(s'|s, a) = \frac{\sum_{n=1}^{N_0} \mathbb{1}_{s^n(s,a)=s'}}{N_0},$$

(8)

where $r^n(s, a)$ denotes the $n$-th observation of the reward, and $c_k^n(s, a)$ denotes the $n$-th observation of the $k$-th cost, and $s^n(s, a)$ denotes the $n$-th observation of the state transition for the state-action pair $(s, a)$, for $n \in [N_0]$. Then, similar to [21], we can use the following LP to obtain an estimate of $V$ (6).

$$\bar{V}_{N_0} = \max \quad (\bar{\boldsymbol{r}}_{N_0})^\top \boldsymbol{q}$$
$$\text{s.t.} \quad \bar{C}_{N_0} \boldsymbol{q} \le \boldsymbol{\alpha} + \lambda_{N_0}$$
$$\bar{B}_{N_0} \boldsymbol{q} \le \boldsymbol{\mu} + \lambda_{N_0}$$
$$\bar{B}_{N_0} \boldsymbol{q} \ge \boldsymbol{\mu} - \lambda_{N_0}$$
$$\boldsymbol{q} \ge 0,$$

(9)

with $\lambda_{N_0}$ being a parameter that we specify later. To bound the estimation gap between $\bar{V}_{N_0}$ and $V$, it is useful to bound the optimal dual solution to (6). To this end, we adopt the approach in [29, 55] that utilizes Slater's condition, which is imposed as an assumption below.

**Assumption 3.1** *There exists a policy $\bar{\pi}$ such that all the resource constraints are satisfied strictly. In other words, there exists an occupancy measure $\bar{\boldsymbol{q}}$ such that $B\bar{\boldsymbol{q}} = \boldsymbol{\mu}$ and $C\bar{\boldsymbol{q}} < \boldsymbol{\alpha}$. In fact, for each state $s \in \mathcal{S}$, there exists a null action that consumes no resource.*

The Slater point $\bar{\boldsymbol{q}}$ can be set as the policy that takes the null action given each state. The estimation error will be related to the gap between the Slater point $\bar{\boldsymbol{q}}$ and the optimal point $\boldsymbol{q}^*$. We then define the lower gap as

$$\text{Gap}_1(N_0, \varepsilon) \ge V - \bar{V}_{N_0}$$

(10)

and the upper gap as

$$\text{Gap}_2(N_0, \varepsilon) \ge \bar{V}_{N_0} - V$$

(11)

with both inequalities (10) and (11) hold with probability at least $1 - \varepsilon$.

### 3.1 Bound the Estimation Gap

We denote by $\text{Rad}(N_0, \varepsilon) = \sqrt{\frac{\log(2/\varepsilon)}{2N_0}}$. Following the standard Hoeffding's inequality, we know that $|\bar{r}_{N_0}(s, a) - \hat{r}(s, a)|$, $|\bar{c}_{k, N_0}(s, a) - \hat{c}_k(s, a)|$, and $|\bar{P}_{N_0}(s'|s, a) - P(s'|s, a)|$ are all upper bounded by $\text{Rad}(N_0, \varepsilon)$ with probability at least $1 - \varepsilon$. We can simply set

$$\text{Gap}_1(N_0, \varepsilon) = \text{Rad}(N_0, \varepsilon)$$

(12)

and

$$\text{Gap}_2(N_0, \varepsilon) = \frac{2\text{Rad}(N_0, \varepsilon)}{\min_{k \in [K]}\{\alpha_k\}} \cdot \left(1 + \frac{|\mathcal{S}|}{1 - \gamma}\right) + \frac{\text{Rad}^2(N_0, \varepsilon)}{\min_{k \in [K]}\{\alpha_k\}} \cdot \left(|\mathcal{S}| + \frac{|\mathcal{S}|^2}{1 - \gamma}\right). \tag{13}$$

We have the following result, where the proof is relegated to appendix.

**Lemma 3.2** *As long as $\lambda_{N_0} = \text{Rad}(N_0, \varepsilon)$, the following inequality*

$$V \leq \bar{V}_{N_0} + \text{Gap}_1(N_0, \varepsilon) \leq V + \text{Gap}_1(N_0, \varepsilon) + \text{Gap}_2(N_0, \varepsilon). \tag{14}$$

*holds with probability at least $1 - (K|\mathcal{S}||\mathcal{A}| - |\mathcal{S}|^2|\mathcal{A}|) \cdot \varepsilon$, where $\text{Gap}_1(N_0, \varepsilon)$ is defined in* (12) *and $\text{Gap}_2(N_0, \varepsilon)$ is defined in* (13).

## 3.2 Characterize One Optimal Basis

We now describe how to identify one optimal basis of the LP (6) as required in Lemma 2.1, by sequentially discarding the sub-optimal actions and the redundant constraints. The formal algorithm to identify such non-zero elements and the constraints is given in Algorithm 1.

---

**Algorithm 1** Algorithm for identifying one optimal basis

---

1: **Input:** the historical sample set $\mathcal{F}_{N_0}$ that contains $N_0$ samples for each $(s, a) \in \mathcal{S} \times \mathcal{A}$.
2: Compute the value of $\bar{V}_{N_0}$ as in (9).
3: Initialize $\mathcal{I}$ to be the whole index set that contains every column index of matrix $B$ in (6) and $\mathcal{J} = [K]$.
4: **for** $i \in \mathcal{I}$ **do**
5:    Let $\mathcal{I}' = \mathcal{I}\backslash\{i\}$.
6:    Compute the value of $\bar{V}_{\mathcal{I}', N_0}$ as in (15).
7:    If $|\bar{V}_{\mathcal{I}', N_0} - \bar{V}_{N_0}| \leq 2\text{Gap}_1(N_0, \varepsilon) + 2\text{Gap}_2(N_0, \varepsilon)$, then we set $\mathcal{I} = \mathcal{I}'$.
8: **end for**
9: **for** $k = 1, \ldots, K$ **do**
10:    Let $\mathcal{J}' = \mathcal{J}\backslash\{q\}$.
11:    Compute the value of $\text{Dual}_{\mathcal{J}', \mathcal{I}, N_0}$ as in (18).
12:    If $|\bar{V}_{N_0} - \text{Dual}_{\mathcal{J}', \mathcal{I}, N_0}| \leq 2\text{Gap}_1(N_0, \varepsilon) + 2\text{Gap}_2(N_0, \varepsilon)$, then we set $\mathcal{J} = \mathcal{J}'$.
13: **end for**
14: **Output:** the set of indexes $\mathcal{I}$ and $\mathcal{J}$.

---

We now explain the intuition why Algorithm 1 works. Denote by $\boldsymbol{q}^*$ an optimal solution to LP (6). We describe how to identify the non-zero elements in $\boldsymbol{q}^*$ and how to identify the constraints such that the values of the non-zero elements of $\boldsymbol{q}^*$ can be uniquely determined by the corresponding linear equation. For each $i$-th element of $\boldsymbol{q}^*$, we compare the value of $V$ (6) against $V$ with an additional constraint that $q_i = 0$. If the two values are different, we identify a non-zero element. To this end, for an index set $\mathcal{I}$, we define an LP, as well as its estimate, as follows.

$$
\begin{array}{llll}
V_{\mathcal{I}} = \max & \hat{\boldsymbol{r}}^\top \boldsymbol{q} & \bar{V}_{\mathcal{I}, N_0} = & \max & (\bar{\boldsymbol{r}}_{N_0})^\top \boldsymbol{q} \\
\text{s.t.} & C\boldsymbol{q} \leq \boldsymbol{\alpha} & & \text{s.t.} & \bar{C}_{N_0}\boldsymbol{q} \leq \boldsymbol{\alpha} + \lambda_{N_0} \\
& B\boldsymbol{q} = \boldsymbol{\mu} & & & |\bar{B}_{N_0}\boldsymbol{q} - \boldsymbol{\mu}| \leq \lambda_{N_0} \\
& \boldsymbol{q}_{\mathcal{I}^c} = 0 & & & \boldsymbol{q}_{\mathcal{I}^c} = 0 \\
& \boldsymbol{q} \geq 0, & & & \boldsymbol{q} \geq 0.
\end{array}
\tag{15}
$$

where $\mathcal{I}^c$ denotes the complementary set of $\mathcal{I}$. Note that if $V - V_{\mathcal{I}} > 0$, we know that $\mathcal{I}^c$ contains a non-zero basic variable. The steps 4-8 in Algorithm 1 reflect this point. Starting from $\mathcal{I}$ denoting the whole index set, we sequentially delete one element $i$ (denoting $(s, a)$ in the infinite horizon discounted problem) from the set $\mathcal{I}$. Once we detected that $V - V_{\mathcal{I}\backslash\{i\}} > 0$, we know that $i$ is a non-zero basic variable and we add $i$ back into the set $\mathcal{I}$. In this way, we can classify all the basic variables into the set $\mathcal{I}$. However, since we do not know the exact value of $V$ and $V_{\mathcal{I}}$, we use the estimates and compare the value of $\bar{V}_{N_0}$ and $\bar{V}_{\mathcal{I}, N_0}$. For this comparison to be valid, the estimation error has to be smaller than the intrinsic gap between $V$ and $V_{\mathcal{I}}$. We define a constant

$$\Delta_1 = \min_{\mathcal{I}}\{V - V_{\mathcal{I}} : V - V_{\mathcal{I}} > 0\} \tag{16}$$

and we need $N_0$ to be large enough such that the estimation gap is smaller than $\Delta_1/2$.

To find the corresponding active constraints, we consider the dual program of $V_{\mathcal{I}}$, where $\mathcal{I}$ is determined in steps 4-8 in Algorithm 1, and similarly, we test which dual variable can be set to $0$ without influencing the dual objective value. For a dual variable index subset $\mathcal{J} \subset [K]$, we consider the dual program as follows.

$$\text{Dual}_{\mathcal{J},\mathcal{I}} = \min \quad \boldsymbol{\alpha}^\top \boldsymbol{y} + \boldsymbol{\mu}^\top \boldsymbol{z} \tag{17a}$$

$$\text{s.t.} \ \ (C(:,\mathcal{I}))^\top \boldsymbol{y} + (B(:,\mathcal{I}))^\top \boldsymbol{z} \geq \hat{\boldsymbol{r}}_{\mathcal{I}} \tag{17b}$$

$$\boldsymbol{y}_{\mathcal{J}^c} = 0 \tag{17c}$$

$$\boldsymbol{y} \geq 0, \boldsymbol{z} \geq -\infty, \tag{17d}$$

where $\mathcal{J}^c = [K]\backslash\mathcal{J}$. The estimate of $\text{Dual}_{\mathcal{J},\mathcal{I}}$, can be obtained from the estimate of its dual, which is given below.

$$\text{Dual}_{\mathcal{J},\mathcal{I},N_0} = \bar{V}_{\mathcal{J},\mathcal{I},N_0} = \max \quad (\bar{\boldsymbol{r}}_{N_0})^\top \boldsymbol{q} \tag{18a}$$

$$\text{s.t.} \ \ \bar{C}_{N_0}(\mathcal{J},:)\boldsymbol{q} \leq \boldsymbol{\alpha} + \lambda_{N_0} \tag{18b}$$

$$|\bar{B}_{N_0}\boldsymbol{q} - \boldsymbol{\mu}| \leq \lambda_{N_0} \tag{18c}$$

$$\boldsymbol{q}_{\mathcal{I}} = 0 \tag{18d}$$

$$\boldsymbol{q} \geq 0, \tag{18e}$$

Similarly, we compare the value of $\text{Dual}_{\mathcal{I}}$, where $\text{Dual}_{\mathcal{I}} = \text{Dual}_{\mathcal{J}',\mathcal{I}}$ with $\mathcal{J}' = [K]$, and $\text{Dual}_{\mathcal{J},\mathcal{I}}$. However, we can only compare the value of their estimates $\text{Dual}_{\mathcal{I},N_0}$ and $\text{Dual}_{\mathcal{J},\mathcal{I},N_0}$. To this end, we define a constant

$$\Delta_2 = \min_{\mathcal{J} \subset [K]} \{\text{Dual}_{\mathcal{J},\mathcal{I}} - \text{Dual}_{\mathcal{I}} : \text{Dual}_{\mathcal{J},\mathcal{I}} - \text{Dual}_{\mathcal{I}} > 0\}. \tag{19}$$

For the comparison to be valid, we need $N_0$ to be large enough such that the estimation error is smaller than $\Delta_2/2$. In this way, we identify the linearly independent binding constraints corresponding to $\boldsymbol{q}^*$, as described in steps 9-13 of Algorithm 1.

## 4 Our Final Algorithm

We now describe our formal algorithm. From the output of Algorithm 1, we characterize one optimal solution. If the sample size $n$ is used in Algorithm 1, we denote by $\mathcal{I}_n$ and $\mathcal{J}_n$ the output of Algorithm 1. To be specific, we have $\boldsymbol{q}^*_{\mathcal{I}_n^c} = 0$ and the non-zero elements $\boldsymbol{q}^*_{\mathcal{I}_n}$ can be given as the solution to

$$\begin{bmatrix} C(\mathcal{J}_n,\mathcal{I}_n) \\ B(:,\mathcal{I}_n) \end{bmatrix} \cdot \boldsymbol{q}^*_{\mathcal{I}_n} = \begin{bmatrix} \boldsymbol{\alpha}_{\mathcal{J}_n} \\ (1-\gamma) \cdot \boldsymbol{\mu} \end{bmatrix}. \tag{20}$$

However, in practice, both the matrices $C(\mathcal{J}_n,\mathcal{I}_n)$ and $B(:,\mathcal{I}_n)$ are unknown. We aim to use the samples to learn the matrices $C(\mathcal{J}_n,\mathcal{I}_n)$ and $B(:,\mathcal{I}_n)$ such that the $\boldsymbol{q}^*_{\mathcal{I}_n}$ can also be determined. Our formal algorithm is given in Algorithm 2. The steps 3-8 in Algorithm 2 is to use Algorithm 1 as a subroutine to identify the set $\mathcal{I}$ and $\mathcal{J}$ that satisfy the conditions in Theorem 2.1. We can show that as long as $n \geq N_0'$, where $N_0'$ is a threshold that depends on the problem parameters, Algorithm 1 correctly obtains the set $\mathcal{I}$ and $\mathcal{J}$ satisfying the conditions in Theorem 2.1. Therefore, we exponentially increase the value of $n$ as input to Algorithm 1 to reach $N_0'$.

A crucial element in Algorithm 2 (step 10) is that we adaptively update the value of $\boldsymbol{\alpha}^n_{\mathcal{J}_{n-1}}$ and $\boldsymbol{\mu}^n(s')$ as in (22) and (23). We then use the updated $\boldsymbol{\alpha}^n_{\mathcal{J}_{n-1}}$ and $\boldsymbol{\mu}^n(s')$ to obtain the value of $\tilde{\boldsymbol{q}}^n_{\mathcal{I}_n}$ as in (21). Such an algorithmic design follows the resolving idea from online LP to achieve a problem-dependent bound. In step 11, we further project $\tilde{\boldsymbol{q}}^{n+1}$ to the set $\{\boldsymbol{q} \geq 0 : \|\boldsymbol{q}\|_1 \leq 2\}$ to obtain $\boldsymbol{q}^{n+1}$. This prevents $\tilde{\boldsymbol{q}}^n$ from behaving ill when $n$ is not large and the estimates of $C(\mathcal{J}_n,\mathcal{I}_n)$ and $B(:,\mathcal{I}_n)$ are not accurate enough. We can show that when $n$ is large enough, $\tilde{\boldsymbol{q}}^{n+1}$ automatically belongs to the set $\{\boldsymbol{q} \geq 0 : \|\boldsymbol{q}\|_1 \leq 2\}$.

## 5 Theoretical Analysis

In this section, we conduct our theoretical analysis. The analysis can be divided into two parts. In the first part, we show that Algorithm 1 can successfully help us identify one optimal basis to work

---

**Algorithm 2** The Adaptive-resolving Algorithm

---

1: **Input:** the number of samples $N$ for each $(s,a) \in \mathcal{S} \times \mathcal{A}$.
2: Initialize $\mathcal{F}_1 = \emptyset$, $\boldsymbol{\alpha}^1 = N \cdot \boldsymbol{\alpha}$ and $\boldsymbol{\mu}^1 = N \cdot \boldsymbol{\mu}$.
3: **for** $n = 1, \ldots, N$ **do**
4:      **if** $n = 2^m$ for an integer $m$ **then**
5:          Obtain the output $\mathcal{I}_n$ and $\mathcal{J}_n$ from Algorithm 1 with the input $\mathcal{F}_n$.
6:      **else**
7:          set $\mathcal{I}_n = \mathcal{I}_{n-1}$ and $\mathcal{J}_n = \mathcal{J}_{n-1}$.
8:      **end if**
9:      Construct estimates $\bar{C}^n(\mathcal{J}_n, \mathcal{I}_n)$ and $\bar{B}^n(:, \mathcal{I}_n)$ using the sample set $\mathcal{F}_n$.
10:      Construct a solution $\tilde{\boldsymbol{q}}^n$ such that $\tilde{\boldsymbol{q}}^n_{\mathcal{I}^c_n} = 0$ and $\tilde{\boldsymbol{q}}^*_{\mathcal{I}_n}$ is the solution to

$$\begin{bmatrix} \bar{C}^n(\mathcal{J}_n, \mathcal{I}_n) \\ \bar{B}^n(:, \mathcal{I}_n) \end{bmatrix} \cdot \tilde{\boldsymbol{q}}^n_{\mathcal{I}_n} = \begin{bmatrix} \dfrac{\boldsymbol{\alpha}^n_{\mathcal{J}_n}}{N - n + 1} \\ \dfrac{\boldsymbol{\mu}^n}{N - n + 1} \end{bmatrix}. \tag{21}$$

11:      Project $\tilde{\boldsymbol{q}}^n$ to the set $\{\boldsymbol{q} : \|\boldsymbol{q}\|_1 \leq 2\}$ to obtain $\boldsymbol{q}^n$.
12:      For each $(s,a) \in \mathcal{I}_n \subset \mathcal{S} \times \mathcal{A}$, we query the model $\mathcal{M}$ to obtain a sample of the reward $r^n(s,a)$ and the costs $c^n_k(s,a)$ for each $k \in \mathcal{J}_n \subset [K]$, as well as the state transition $s^n(s,a)$.
13:      Update $\mathcal{F}_{n+1} = \mathcal{F}_n \cup \{r^n(s,a), c^n_k(s,a), s^n(s,a), \forall (s,a) \in \mathcal{S} \times \mathcal{A}, \forall k \in [K]\}$.
14:      Denote by $\boldsymbol{c}^n(s,a) = (c^n_k(s,a))_{\forall k \in \mathcal{J}_n}$ and do the update:

$$\boldsymbol{\alpha}^{n+1}_{\mathcal{J}_n} = \boldsymbol{\alpha}^n_{\mathcal{J}_n} - \sum_{(s,a) \in \mathcal{I}_n} \boldsymbol{c}^n(s,a) \cdot q^n(s,a). \tag{22}$$

15:      Do the update:

$$\boldsymbol{\mu}^{n+1}(s') = \boldsymbol{\mu}^n(s') - \sum_{(s,a) \in \mathcal{I}} q^n(s,a) \cdot (\delta_{s',s} - \gamma \mathbb{1}_{\{s' = s^n(s,a)\}}), \quad \forall s' \in \mathcal{S}. \tag{23}$$

     where $\mathbb{1}_{\{s' = s^n(s,a)\}}$ is an indicator function of whether the state transition $s^n(s,a)$ equals $s'$.
16: **end for**
17: We define $\bar{\boldsymbol{q}}^N$ such that $\bar{\boldsymbol{q}}^N_{\mathcal{I}^c_N} = 0$ and $\bar{\boldsymbol{q}}^N_{\mathcal{I}_N} = \frac{1}{N} \cdot \sum_{n=1}^N \boldsymbol{q}^n_{\mathcal{I}_N}$. We then define a policy $\bar{\pi}^N$

$$P(\bar{\pi}^N(s) = a) = \begin{cases} \dfrac{\bar{q}^N(s,a)}{\sum_{a' \in \mathcal{A}} \bar{q}^N(s,a')}, & \text{if } \displaystyle\sum_{a' \in \mathcal{A}} \bar{q}^N(s,a') > 0 \\ 1/|\mathcal{A}|, & \text{if } \displaystyle\sum_{a' \in \mathcal{A}} \bar{q}^N(s,a') = 0. \end{cases} \tag{24}$$

18: **Output:** policy $\bar{\pi}^N$.

---

with. In the second part, we show how to learn the optimal distribution over the optimal basis we have identified.

We now present the theorem showing that Algorithm 1 indeed helps us identify one optimal basis with a high probability. In practice, the value of $\varepsilon$ will be set to $1/N$ in the following theorem.

**Theorem 5.1** *For any $\varepsilon > 0$, as long as $N_0$ satisfies the condition*

$$2\text{Gap}_1(N_0, \varepsilon) + 2\text{Gap}_2(N_0, \varepsilon) \leq \min\{\Delta_1, \Delta_2\} \tag{25}$$

*the outputs $\mathcal{I}_{N_0}$ and $\mathcal{J}_{N_0}$ of Algorithm 1 satisfy the conditions described in Lemma 2.1 with probability at least $1 - (K|\mathcal{S}||\mathcal{A}| - |\mathcal{S}|^2|\mathcal{A}|) \cdot \varepsilon$. Moreover, the sets $\mathcal{I}_n$ and $\mathcal{J}_n$ will be common for any $n \geq N_0$ satisfying (25), which we denote by $\mathcal{I}^*$ and $\mathcal{J}^*$.*

An important problem parameter related to $\mathcal{I}^*$ and $\mathcal{J}^*$ can be described as follows. Define

$$A^* = \begin{bmatrix} C(\mathcal{J}^*, \mathcal{I}^*) \\ B(:, \mathcal{I}^*) \end{bmatrix}. \tag{26}$$

We then denote by $\{\sigma_1(A^*), \ldots, \sigma_{|\mathcal{S}|+K'}(A^*)\}$ the eigenvalues of the matrix $A^*$. We define $\sigma$ as

$$\sigma = \min\left\{|\sigma_1(A^*)|, \ldots, |\sigma_{|\mathcal{S}|+K'}(A^*)|\right\}. \tag{27}$$

From the non-singularity of the matrix $A^*$, we know that $\sigma > 0$. We then have the following bound.

**Theorem 5.2** *With a sample complexity bound of*

$$O\left(\frac{(|\mathcal{S}|+K)^3 \cdot |\mathcal{S}| \cdot |\mathcal{A}|}{\alpha^2 \cdot \xi \cdot \sigma(1-\gamma) \cdot \min\{\sigma^2, (1-\gamma)^2 \cdot \Delta\}} \cdot \frac{\log^2(1/\varepsilon)}{\varepsilon}\right),$$

*where $\Delta = \min\{\Delta_1^2, \Delta_2^2\}$, $\xi = \min_{(s,a)\in\mathcal{I}^*}\{q^*(s,a)\}$ and $\boldsymbol{q}^*$ denotes the optimal soluton to LP* (6)
*corresponding to the optimal basis $\mathcal{I}^*$ and $\mathcal{J}^*$, we obtain a policy $\bar{\pi}^N$ from Algorithm* 2 *(defined in*
(24)*) such that*

$$V_r(\pi^*, \mu_1) - V_r(\bar{\pi}^N, \mu_1) \leq \varepsilon \ \text{ and } \ V_k(\bar{\pi}^N, \mu_1) - \alpha_k \leq \varepsilon, \ \forall k \in [K].$$

Our Algorithm 2 can be directly applied to solving MDP problems without resource constraints and our bounds in Theorem E.2 and Theorem 5.2 still hold. Note that in the MDP problems, the parameter $\sigma$ can be lower bounded by $1 - \gamma$, which follows from the fact that the matrix $A^*$ can simply be represented by the probability transition matrix. Then, we have the following sample complexity bound for our Algorithm 2.

**Proposition 5.3** *For the MDP problems without resource constraints, i.e., $K = 0$, with a sample complexity bound of*

$$O\left(\frac{|\mathcal{S}|^4 \cdot |\mathcal{A}|}{(1-\gamma)^4 \cdot \xi \cdot \Delta} \cdot \frac{\log^2(1/\varepsilon)}{\varepsilon}\right), \tag{28}$$

*we obtain a policy $\bar{\pi}^N$ from Algorithm* 2 *(defined in* (24)*) such that*

$$V_r(\pi^*, \mu_1) - V_r(\bar{\pi}^N, \mu_1) \leq \varepsilon \ \text{ and } \ V_k(\bar{\pi}^N, \mu_1) - \alpha_k \leq \varepsilon, \ \forall k \in [K].$$

In terms of the dependency of our sample complexity bound on other problem parameters such as $|\mathcal{S}|$, $|\mathcal{A}|$, and $1 - \gamma$, we compare to the series of work [58, 67, 68, 2, 27], that subsequently achieves a sample complexity bound of $O\left(\frac{|\mathcal{S}| \cdot |\mathcal{A}|}{(1-\gamma)^3 \cdot \varepsilon^2}\right)$, where the dependency over $|\mathcal{S}|$, $|\mathcal{A}|$, and $1 - \gamma$ is optimal [24, 43]. Our sample complexity bound in (28) has a worse dependency in terms of $|\mathcal{S}|$ and $1 - \gamma$. This is because we construct an empirical LP to estimate the value of LP (6) and resolve the linear equation as in (21), where the size of the LP (which is $|\mathcal{S}|$) and the eigenvalues of the matrix $A^*$ (which is bounded by $1 - \gamma$) will play a part. However, our bound (28) enjoys a better dependency in terms of $\varepsilon$.

## 6  Conclusions

In this paper, we develop the first instance-dependent $\tilde{O}(1/\epsilon)$ sample complexity for constrained MDP problems. We characterize the instance hardness via corner points of the LP formulation and we develop a resolving algorithm to learn the optimal solution while sticking to the identified optimal basis. The work presented by this paper advances the field of Machine Learning and the algorithmic ideas developed in this paper have a broader impact to inspire new algorithms. Our results are developed for the tabular settings, which pose some limitations to the real-world applications of our methods. We leave the extensions to more involved settings for future work.

## Acknoledgement

Jiashuo Jiang is generously supported by the early career scheme 26210223 and the general research fund 16204024 from Research Grants Council, Hong Kong.

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

## A    Numerical Experiments

We implement our Algorithm 2 to study the numerical performance. We consider a CMDP problem with the state space $|\mathcal{S}| = 10$ and the action space $|\mathcal{A}| = 10$. We set the discount factor $\gamma = 0.7$. We then randomly generate the probability transition kernel $P$. To be specific, for each state $s \in \mathcal{S}$, action $a \in \mathcal{A}$, and the future state $s' \in \mathcal{S}$, we uniformly generate a randomly variable $p_{s,a,s'}$. Then, the transition probability is defined as $P(s'|s,a) = \frac{p_{s,a,s'}}{\sum_{s'' \in \mathcal{S}} p_{s,a,s''}}$. For each state-action pair $(s,a) \in \mathcal{S} \times \mathcal{A}$, the expected reward $\hat{r}(s,a)$ is uniformly generated from the interval $[1,2]$ (with the reward for the first action set to be 0). The actual reward $r(s,a) = \hat{r}(s,a) + \eta$, where $\eta$ is uniformly distributed among $[-0.5, 0.5]$. There are $K = 5$ constraints and for each constraint $k \in [K]$ and each state-action pair $(s,a) \in \mathcal{S} \times \mathcal{A}$, we define the expected cost $\hat{c}_k(s,a)$ to be uniformly generated from $[1,2]$. The actual cost $c_k(s,a) = \hat{c}_k(s,a) + \eta'$, where $\eta'$ is uniformly distributed among $[-0.5, 0.5]$.

For each total iterations $N$, We apply Algorithm 2 and obtain the output $q^1, \ldots, q^N$. We compare $\bar{q}^N$ with the optimal occupancy measure. Since our algorithm is a randomized algorithm, we study the performance of our algorithm in expectation. To be specific, given the problem instance and a fixed $N$, we implement our algorithm repeatedly for $M = 500$ rounds. Denote by $\bar{q}_m^N$ the output of our Algorithm 2 at round $m$, for $m \in [M]$. We define the error term as $\text{Err}(N) = \frac{1}{M} \cdot \sum_{m=1}^{M} \|\bar{q}_m^N - q^*\|_1 / \|q^*\|_1$. We study how the error term $\text{Err}(N)$ scales with $N$. The results are displayed in Figure 2. As we can see, the error term $\text{Err}(N)$ decreases quickly with respect to $N$. Moreover, since our Algorithm 2 requires only solving a set of linear equations in each iteration, the computation cost of our Algorithm 2 is also moderate.

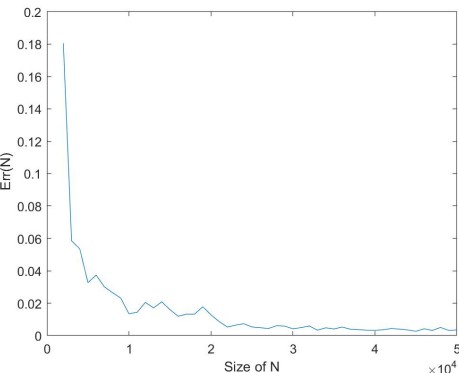

Figure 2: The computational performance of Algorithm 2. The x-label denotes the size of $N$, while the y-label denotes the error term $\text{Err}(N)$.

## B    Proof of Lemma 2.1

Let $q^*$ be a basic optimal solution to LP (6). We also denote by $\mathcal{I}$ the index set of $q^*$ such that $q_{\mathcal{I}}^* > 0$. We then denote by $\mathcal{J}' \in [K]$ the set of active constraints at $q^*$. Then, the following linear equations must be satisfied by $q^*$.

$$C(\mathcal{J}', \mathcal{I})q_{\mathcal{I}}^* = \alpha_{\mathcal{J}'} \quad \text{and} \quad B(:,\mathcal{I})q_{\mathcal{I}}^* = \mu. \tag{29}$$

From the property of the basic optimal solution, we must have $|\mathcal{J}'| + b \geq |\mathcal{I}|$.

If $|\mathcal{J}'| + b = |\mathcal{I}|$, then we know that we already find the desired index set $\mathcal{I}$ and $\mathcal{J} = \mathcal{J}'$.

Otherwise, if $|\mathcal{J}'| + b > |\mathcal{I}|$, then we know that the LP (6) is degenerate. However, now the linear system (29) is over-determined, i.e., there must be $|\mathcal{J}'| + b - |\mathcal{I}|$ number of equations can be implied by the others. It only remains to show that those redundant equations are all in the set $\mathcal{J}'$. This step can be done can showing that the matrix $B(:,\mathcal{I})$ has a full row rank. We prove this by showing contradiction.

For the infinite horizon discounted problem, suppose that $B(:, \mathcal{I})$ does not have full row rank, there must exists $\boldsymbol{\beta} \in \mathbb{R}^{|\mathcal{S}|}$ such that

$$\boldsymbol{\beta}^\top B(:, \mathcal{I}) = \mathbf{0}. \tag{30}$$

On the other hand, denote by $\pi^*$ the optimal policy corresponding to $\boldsymbol{q}^*$. We then define a matrix $\Gamma \in \mathbb{R}^{|\mathcal{I}| \times \mathcal{S}}$ such that the $s$-th column of $\Gamma$ is a vector that takes 0 for all the $(s', a) \in \mathcal{I}$-th element if $s' \neq s$, and take a value of $\pi^*(a|s)$ for all $a$ such that $(s, a) \in \mathcal{I}$. We know that

$$B(:, \mathcal{I})\Gamma = I - \gamma \cdot P^{\pi^*}$$

with $I \in \mathbb{R}^{|\mathcal{S}| \times |\mathcal{S}|}$ being the identify matrix and $P^{\pi^*}$ being the transition probability matrix under the policy $\pi^*$, with the element at the $s$-th row and $s'$-th column denoting the probability of transiting from state $s$ to state $s'$ under the policy $\pi^*$. It is well known that the matrix $I - \gamma \cdot P^{\pi^*}$ is non-singular (see for example the Gersgorin's Theorem in [30]). Therefore, for any vector $\boldsymbol{\beta} \in \mathbb{R}^{|\mathcal{S}|}$, we know that

$$\boldsymbol{\beta}^\top B(:, \mathcal{I})\Gamma = \boldsymbol{\beta}^\top (I - \gamma \cdot P^{\pi^*}) \neq 0$$

which contradicts with (60). Our proof is thus completed.

## C   Proof of Lemma 3.2

We now condition on the event that $|\bar{r}_{N_0}(s, a) - \hat{r}(s, a)|$, $|\bar{c}_{k,N_0}(s, a) - \hat{c}_k(s, a)|$, and $|\bar{P}_{N_0}(s'|s, a) - P(s'|s, a)|$ for each $k \in [K]$, $(s, a) \in \mathcal{S} \times \mathcal{A}$ and $s' \in \mathcal{S}$ are all bounded by $\mathrm{Rad}(N_0, \varepsilon)$. From the union bound, we know that this event happens with probability at least $1 - (K|\mathcal{S}||\mathcal{A}| - |\mathcal{S}|^2|\mathcal{A}|) \cdot \varepsilon$.

Note that for LP (4), by summing up the constraint (4c) for all $s \in \mathcal{S}$, we obtain that any feasible solution $\boldsymbol{q}$ for LP (4) would satisfy

$$\|\boldsymbol{q}\|_1 = 1. \tag{31}$$

We first upper bound the gap $V - \bar{V}_{N_0}$. Denote by $\boldsymbol{q}^*$ one optimal solution to $V$. Then, from the feasibility of $\boldsymbol{q}^*$, we know that

$$\bar{C}_{N_0}\boldsymbol{q}^* = C\boldsymbol{q}^* + (\bar{C}_{N_0} - C)\boldsymbol{q}^* \leq \boldsymbol{\alpha} + \mathrm{Rad}(N_0, \varepsilon) \leq \boldsymbol{\alpha} + \lambda_{N_0} \tag{32}$$

where the first inequality follows from $\|\boldsymbol{q}^*\|_1 = 1$ (31) and all elements of $\bar{C}_{N_0} - C$ are upper bounded by $\mathrm{Rad}(N_0, \varepsilon)$. Also, we know that

$$\bar{B}_{N_0}\boldsymbol{q}^* = B\boldsymbol{q}^* + (\bar{B}_{N_0} - B)\boldsymbol{q}^* \leq (1 - \gamma) \cdot \boldsymbol{\mu} + \gamma \cdot \mathrm{Rad}(N_0, \varepsilon) \leq (1 - \gamma) \cdot \boldsymbol{\mu} + \lambda_{N_0} \tag{33}$$

where the first inequality follows from $\|\boldsymbol{q}^*\|_1 = 1$ (31) and all elements of $\bar{B}_{N_0} - B$ are upper bounded by $\gamma \cdot \mathrm{Rad}(N_0, \varepsilon)$. Similarly, we have that

$$\bar{B}_{N_0}\boldsymbol{q}^* = B\boldsymbol{q}^* + (\bar{B}_{N_0} - B)\boldsymbol{q}^* \geq (1 - \gamma) \cdot \boldsymbol{\mu} - \gamma \cdot \mathrm{Rad}(N_0, \varepsilon) \geq (1 - \gamma) \cdot \boldsymbol{\mu} - \lambda_{N_0}. \tag{34}$$

Therefore, as long as

$$\lambda_{N_0} \geq \mathrm{Rad}(N_0, \varepsilon), \tag{35}$$

we know that $\boldsymbol{q}^*$ is a feasible solution to $\bar{V}_{N_0}$. We have that

$$\bar{\boldsymbol{r}}_{N_0}^\top \boldsymbol{q}^* \geq \hat{\boldsymbol{r}}^\top \boldsymbol{q}^* - \mathrm{Rad}(N_0, \varepsilon) \tag{36}$$

by noting $\|\boldsymbol{q}^*\|_1 = 1$ (31) and all elements of $\bar{\boldsymbol{r}}_{N_0} - \hat{\boldsymbol{r}}$ are upper bounded by $\mathrm{Rad}(N_0, \varepsilon)$. Therefore, we can obtain the bound

$$V - \mathrm{Rad}(N_0, \varepsilon) \leq \bar{\boldsymbol{r}}_{N_0}^\top \boldsymbol{q}^* \leq \bar{V}_{N_0}. \tag{37}$$

We then lower bound the gap $V - \bar{V}_{N_0}$. We first define

$$
\begin{aligned}
\bar{V}_{N_0}(\lambda'_{N_0}) = \max \quad & (\bar{\boldsymbol{r}}_{N_0} - \lambda'_{N_0})^\top \boldsymbol{q} \\
\text{s.t.} \quad & \bar{C}_{N_0}\boldsymbol{q} \leq \boldsymbol{\alpha} + \lambda_{N_0} \\
& \bar{B}_{N_0}\boldsymbol{q} \leq \boldsymbol{\mu} + \lambda_{N_0} \\
& \bar{B}_{N_0}\boldsymbol{q} \geq \boldsymbol{\mu} - \lambda_{N_0} \\
& \boldsymbol{q} \geq 0,
\end{aligned}
\tag{38}
$$

for any constant $\lambda'_{N_0}$. Clearly, any optimal solution $\bar{q}^*$ to $\bar{V}_{N_0}$ will be a feasible solution to $\bar{V}_{N_0}(\lambda'_{N_0})$. Moreover, by summing up the constraints $\bar{B}_{N_0}q \leq \mu + \lambda_{N_0}$ for all $s \in \mathcal{S}$, we know that

$$\|\bar{q}^*\|_1 \leq 1 + \frac{|\mathcal{S}|}{1-\gamma} \cdot \lambda_{N_0}. \tag{39}$$

Then, it holds that

$$\bar{V}_{N_0} \leq (\bar{r}_{N_0} - \lambda'_{N_0})^\top \bar{q}^* + \lambda'_{N_0} \cdot \|\bar{q}^*\|_1 \leq \bar{V}_{N_0}(\lambda'_{N_0}) + \lambda'_{N_0} + \frac{|\mathcal{S}|}{1-\gamma} \cdot \lambda_{N_0}\lambda'_{N_0}. \tag{40}$$

We then compare the value between $\bar{V}_{N_0}(\lambda'_{N_0})$ and $V$. The dual of $\bar{V}_{N_0}(\lambda'_{N_0})$ is given below.

$$\begin{aligned}
\text{Dual}'_{N_0}(\lambda'_{N_0}) = \min \quad & \boldsymbol{\alpha}^\top \boldsymbol{y} + \boldsymbol{\mu}^\top(\boldsymbol{z}_1 - \boldsymbol{z}_2) + \lambda_{N_0} \cdot (\|\boldsymbol{y}\|_1 + \|\boldsymbol{z}_1\|_1 + \|\boldsymbol{z}_2\|_1) \\
\text{s.t.} \quad & \bar{C}_{N_0}^\top \boldsymbol{y} + \bar{B}_{N_0}^\top(\boldsymbol{z}_1 - \boldsymbol{z}_2) \geq \bar{r}_{N_0} - \lambda'_{N_0} \\
& \boldsymbol{y} \geq 0, \boldsymbol{z}_1 \geq 0, \boldsymbol{z}_2 \geq 0.
\end{aligned} \tag{41}$$

Denote by $\boldsymbol{y}^*$ and $\boldsymbol{z}^*$ one optimal solution to the dual of LP (6), given below.

$$\begin{aligned}
\text{Dual} = \min \quad & \boldsymbol{\alpha}^\top \boldsymbol{y} + \boldsymbol{\mu}^\top \boldsymbol{z} \\
\text{s.t.} \quad & C^\top \boldsymbol{y} + B^\top \boldsymbol{z} \geq \hat{\boldsymbol{r}} \\
& \boldsymbol{y} \geq 0, \boldsymbol{z} \geq -\infty.
\end{aligned} \tag{42}$$

We now show that $\boldsymbol{y}^*$ and $\boldsymbol{z}^*$ is also a feasible solution to $\text{Dual}'_{N_0}$, with $\lambda'_{N_0} = \frac{1}{\min_{k \in [K]}\{\alpha_k\}} \cdot \left(1 + \frac{|\mathcal{S}|}{1-\gamma}\right) \cdot \text{Rad}(N_0, \varepsilon)$. We have the following claim regarding the upper bound on $\|\boldsymbol{y}^*\|_\infty$ and $\|\boldsymbol{z}^*\|_\infty$.

**Claim C.1** *There exists an optimal solution $\boldsymbol{y}^*$ and $\boldsymbol{z}^*$ to the Dual (42) such that*

$$\|\boldsymbol{y}^*\|_1 \leq \frac{1}{\min_{k \in [K]}\{\alpha_k\}} \quad \text{and} \quad \|\boldsymbol{z}^*\|_\infty \leq \frac{1}{1-\gamma} \cdot \frac{1}{\min_{k \in [K]}\{\alpha_k\}}.$$

Then, we define $\bar{\boldsymbol{y}}^* = \boldsymbol{y}^*$, $\boldsymbol{z}_1^* = \max\{0, \boldsymbol{z}^*\}$ and $\boldsymbol{z}_2^* = \max\{0, -\boldsymbol{z}^*\}$. We have

$$\begin{aligned}
\bar{C}_{N_0}^\top \bar{\boldsymbol{y}}^* + \bar{B}_{N_0}^\top(\boldsymbol{z}_1^* - \boldsymbol{z}_2^*) &\geq C^\top \boldsymbol{y}^* + B^\top \boldsymbol{z}^* - \text{Rad}(N_0, \varepsilon) \cdot (\|\boldsymbol{y}^*\|_1 + \|\boldsymbol{z}^*\|_1) \\
&\geq C^\top \boldsymbol{y}^* + B^\top \boldsymbol{z}^* - \text{Rad}(N_0, \varepsilon) \cdot \frac{1}{\min_{k \in [K]}\{\alpha_k\}} \cdot \left(1 + \frac{|\mathcal{S}|}{1-\gamma}\right) \\
&\geq \hat{\boldsymbol{r}} - \frac{1}{\min_{k \in [K]}\{\alpha_k\}} \cdot \left(1 + \frac{|\mathcal{S}|}{1-\gamma}\right) \cdot \text{Rad}(N_0, \varepsilon).
\end{aligned}$$

Thus, we know that $\bar{\boldsymbol{y}}^*$ and $\boldsymbol{z}_1^*, \boldsymbol{z}_2^*$ is also a feasible solution to $\text{Dual}_{N_0}(\lambda'_{N_0})$, and we have

$$\text{Dual}'_{N_0}(\lambda'_{N_0}) \leq \text{Dual} + \lambda_{N_0} \cdot (\|\boldsymbol{y}^*\|_1 + \|\boldsymbol{z}^*\|_1) \leq \text{Dual} + \frac{\text{Rad}(N_0, \varepsilon)}{\min_{k \in [K]}\{\alpha_k\}} \cdot \left(1 + \frac{|\mathcal{S}|}{1-\gamma}\right). \tag{43}$$

Combing (43) with (40) and also noting that $\bar{V}_{N_0}(\lambda'_{N_0}) = \text{Dual}'_{N_0}(\lambda'_{N_0})$, we have

$$\bar{V}_{N_0} \leq V + \frac{2\text{Rad}(N_0, \varepsilon)}{\min_{k \in [K]}\{\alpha_k\}} \cdot \left(1 + \frac{|\mathcal{S}|}{1-\gamma}\right) + \frac{\text{Rad}^2(N_0, \varepsilon)}{\min_{k \in [K]}\{\alpha_k\}} \cdot \left(|\mathcal{S}| + \frac{|\mathcal{S}|^2}{1-\gamma}\right)$$

which completes our proof.

**Proof of Claim C.1.** We first bound $\|\boldsymbol{y}^*\|_\infty$. We utilize the approach in [29, 55]. We define a Lagrangian function, with only $\boldsymbol{y}$ as the Lagrangian dual variable.

$$L(\boldsymbol{y}, \boldsymbol{q}) := \boldsymbol{\alpha}^\top \boldsymbol{y} + \hat{\boldsymbol{r}}^\top \boldsymbol{q} - \boldsymbol{y}^\top C\boldsymbol{q} \tag{44}$$

where the feasible set for $\boldsymbol{q}$ is $\{\boldsymbol{q} \geq 0 : B\boldsymbol{q} = \boldsymbol{\mu}\}$ and the feasible set for $\boldsymbol{y}$ is $\{\boldsymbol{y} \geq 0\}$. Following Lemma 3 in [55], it is without loss of generality to restrict the feasible set of $\boldsymbol{y}$ to the set $\{\boldsymbol{y} \geq 0 : \|\boldsymbol{y}\|_1 \leq \rho\}$, where the constant $\rho$ is defined as

$$\rho = \frac{\hat{\boldsymbol{r}}^\top \boldsymbol{q}^* - \hat{\boldsymbol{r}}^\top \bar{\boldsymbol{q}}}{\min_{k \in [K]}\{\alpha_k - C(k,:)\bar{\boldsymbol{q}}\}} \tag{45}$$

where $\bar{q}$ is the occupancy measure that satisfies Slater's condition as stated in Theorem 3.1. Note that we have

$$\rho \leq \frac{1}{\min_{k \in [K]} \{\alpha_k\}}.$$

Therefore, we obtain the following bound on $\boldsymbol{y}^*$:

$$\|\boldsymbol{y}^*\|_1 \leq \rho. \tag{46}$$

We now proceed to bound $\boldsymbol{z}^*$. Denote by $\boldsymbol{q}^*$ the optimal solution corresponding to the optimal dual solution $(\boldsymbol{y}^*, \boldsymbol{z}^*)$, with $\boldsymbol{y}^*$ bounded as in (46). We also denote by $\pi^*$ the optimal policy corresponding to $\boldsymbol{q}^*$. Then, from the complementary slackness condition, as long as $\boldsymbol{q}^*(s,a) > 0$ for a $(s,a)$, we must have

$$C(:,(s,a))^\top \boldsymbol{y}^* + B(:,(s,a))^\top \boldsymbol{z}^* = \hat{r}(s,a). \tag{47}$$

We now multiply both sides of (47) by $q^*(s,a)/\sum_{a' \in \mathcal{A}} q^*(s,a')$, and sum up over $a$, for each state $s$. Then we get

$$(C^{\pi^*})^\top \boldsymbol{y}^* + B^{\pi^*} \boldsymbol{z}^* = \hat{\boldsymbol{r}}^{\pi^*}. \tag{48}$$

Here, $C^{\pi^*} \in \mathbb{R}^{K \times |\mathcal{S}|}$ and the element at $k$-th row and $s$-th column is $\sum_{a \in \mathcal{A}} c_k(s,a) \cdot \frac{q^*(s,a)}{\sum_{a' \in \mathcal{A}} q^*(s,a')}$. $B^{\pi^*} = I - \gamma \cdot P^{\pi^*} \in \mathbb{R}^{|\mathcal{S}| \times \mathcal{S}}$, where $P^{\pi^*}(s,s') = \sum_{a \in \mathcal{A}} \pi^*(a|s) \cdot P(s'|s,a)$ denotes the transition probability matrix under the policy $\pi^*$. Also, $\hat{\boldsymbol{r}}^{\pi^*} \in \mathbb{R}^{|\mathcal{S}|}$ with $\hat{r}^{\pi^*}(s) = \sum_{a \in \mathcal{A}} \hat{r}(s,a) \cdot \frac{q^*(s,a)}{\sum_{a' \in \mathcal{A}} q^*(s,a')}$. Then, we have

$$\boldsymbol{z}^* = (B^{\pi^*})^{-1} \cdot \left( \hat{\boldsymbol{r}}^{\pi^*} - (C^{\pi^*})^\top \boldsymbol{y}^* \right) \tag{49}$$

From [37], we know that

$$\|(B^{\pi^*})^{-1}\|_\infty \leq \frac{1}{1-\gamma}. \tag{50}$$

Also, from the bound on $\boldsymbol{y}^*$ in (46), we know that

$$\left\| \hat{\boldsymbol{r}}^{\pi^*} - (C^{\pi^*})^\top \boldsymbol{y}^* \right\|_\infty \leq \rho. \tag{51}$$

Therefore, we have that

$$\|\boldsymbol{z}^*\|_\infty \leq \frac{\rho}{1-\gamma}, \tag{52}$$

which completes our proof. $\qquad \square$

## D Proof of Theorem 5.1

We now condition on the event that $|\bar{r}_{N_0}(s,a) - \hat{r}(s,a)|$, $|\bar{c}_{k,N_0}(s,a) - \hat{c}_k(s,a)|$, and $|\bar{P}_{N_0}(s'|s,a) - P(s'|s,a)|$ for each $k \in [K]$, $(s,a) \in \mathcal{S} \times \mathcal{A}$ and $s' \in \mathcal{S}$ are all bounded by $\mathrm{Rad}(N_0, \varepsilon)$. We know that this event happens with probability at least $1 - (K|\mathcal{S}||\mathcal{A}| - |\mathcal{S}|^2|\mathcal{A}|) \cdot \varepsilon$.

Following the procedure in Algorithm 1, we identify an index set $\mathcal{I}$, where we set $\boldsymbol{q}_i = 0$ for each $i \in \mathcal{I}^c$. Note that we cannot further delete one more $i$ from the set $\mathcal{I}$. Otherwise, when deleting this particular $i$ in $\mathcal{I}$ by setting $\mathcal{I}' = \mathcal{I} \backslash \{i\}$, we will have $V_{\mathcal{I}'} = V$ and as a result of Lemma 3.2, we have

$$|\bar{V}_{N_0} - \bar{V}_{\mathcal{I}',N_0}| \leq |V - V_{\mathcal{I}'}| + 2\mathrm{Gap}_1(N_0, \varepsilon) + 2\mathrm{Gap}_2(N_0, \varepsilon) = 2\mathrm{Gap}_1(N_0, \varepsilon) + 2\mathrm{Gap}_2(N_0, \varepsilon) \tag{53}$$

Therefore, if it is feasible (objective value does not change) to delete one more $i$ from the index set $\mathcal{I}$, our algorithm will already do so.

We denote by $\boldsymbol{q}^*$ the optimal solution to LP (6), corresponding to the optimal basis $\mathcal{I}$ identified by Algorithm 1. We know that $\boldsymbol{q}_{\mathcal{I}}^* > 0$ and $\boldsymbol{q}_{\mathcal{I}^c}^* = 0$, which implies that

$$\begin{aligned} V = V_{\mathcal{I}} = \max \quad & (\hat{\boldsymbol{r}}_{\mathcal{I}})^\top \boldsymbol{q} \\ \text{s.t.} \quad & C(:,\mathcal{I})\boldsymbol{q} \leq \boldsymbol{\alpha} \\ & B(:,\mathcal{I})\boldsymbol{q} = \boldsymbol{\mu} \\ & \boldsymbol{q} \geq 0, \end{aligned} \tag{54}$$

Note that in the formulation of (54), we simply discard the columns of the constraint matrix in the index set $\mathcal{I}$. Thus, one optimal solution to (54) will just be $q_{\mathcal{I}}^*$. The dual of (54) is

$$\begin{aligned}
\text{Dual}_{\mathcal{I}} = \min \ \ & \boldsymbol{\alpha}^\top \boldsymbol{y} + \boldsymbol{\mu}^\top \boldsymbol{z} \\
\text{s.t. } \ \ & (C(:,\mathcal{I}))^\top \boldsymbol{y} + (B(:,\mathcal{I}))^\top \boldsymbol{z} \geq \hat{\boldsymbol{r}}(\mathcal{I}) \\
& \boldsymbol{y} \geq 0, \boldsymbol{z} \geq -\infty,
\end{aligned} \tag{55}$$

From the complementary slackness condition, we know that for the optimal dual variable corresponding to $q_{\mathcal{I}}^*$, all the constraints in (55) must hold with equality. Therefore, we have the following result.

**Claim D.1** *It holds that*

$$\begin{aligned}
V = V_{\mathcal{I}} = \text{Dual}_{\mathcal{I}} = \min \ \ & \boldsymbol{\alpha}^\top \boldsymbol{y} + \boldsymbol{\mu}^\top \boldsymbol{z} \\
\text{s.t. } \ \ & (C(:,\mathcal{I}))^\top \boldsymbol{y} + (B(:,\mathcal{I}))^\top \boldsymbol{z} = \hat{\boldsymbol{r}}(\mathcal{I}) \\
& \boldsymbol{y} \geq 0, \boldsymbol{z} \geq -\infty.
\end{aligned} \tag{56}$$

The proof of the claim is relegated to the end of this proof. We now show that Algorithm 1 identify the linearly independent binding constraints for $q^*$ such that the conditions in Lemma 2.1 are satisfied.

Denote by $C(\mathcal{J},\mathcal{I})$ a sub-matrix of $C$ such that the rows in the index set $\mathcal{J} \subset [K]$ and the columns in the index set $\mathcal{I}$ of the matrix $C$ remain. Now if the matrix

$$A = [C(\mathcal{J},\mathcal{I}); B(:,\mathcal{I})] \tag{57}$$

is singular, there must be

(i). one row in $C(\mathcal{J},\mathcal{I})$ can be expressed as the linear combination of the other rows in $A$ (we know that the matrix $B(:,\mathcal{I})$ has full row rank from Lemma 2.1);

(ii). or one column in $A$ can be expressed as the linear combination of other columns.

We consider the situation (i). Denote this row as $k'$ and we know that restricting $y_{k'} = 0$ will not change the objective value of $\text{Dual}_{\mathcal{I}}$, which implies that $k'$ will be added to the index set $\mathcal{J}$. This is because $y_{k'}$ cannot be a basic variable, otherwise, the optimal basis of the Dual (56) will be linearly dependent. Therefore, we obtain a contradiction and we know that the matrix $A$ must have full row rank. Thus, situation (i) will not happen.

We then consider the situation (ii). Note that we have

$$\begin{aligned}
V = \text{Dual}_{\mathcal{J},\mathcal{I}} = \min \ \ & \boldsymbol{\alpha}_{\mathcal{J}}^\top \boldsymbol{y} + \boldsymbol{\mu}^\top \boldsymbol{z} \\
\text{s.t. } \ \ & (C(\mathcal{J},\mathcal{I}))^\top \boldsymbol{y} + (B(:,\mathcal{I}))^\top \boldsymbol{z} \geq \hat{\boldsymbol{r}}_{\mathcal{I}} \\
& \boldsymbol{y} \geq 0, \boldsymbol{z} \geq -\infty,
\end{aligned} \tag{58}$$

Comparing the formulation (58) to the formulation (55), we only remain the rows of the matrix $C(:,\mathcal{I})$ in the index set $\mathcal{J}$, where it is feasible (objective value does not change) to set $\boldsymbol{y}_{\mathcal{J}^c} = 0$. The dual of (58) is

$$\begin{aligned}
V = V_{\mathcal{J},\mathcal{I}} = \max \ \ & (\hat{\boldsymbol{r}}_{\mathcal{I}})^\top \boldsymbol{q} \\
\text{s.t. } \ \ & C(\mathcal{J},\mathcal{I})\boldsymbol{q} \leq \boldsymbol{\alpha}_{\mathcal{J}} \\
& B(:,\mathcal{I})\boldsymbol{q} = \boldsymbol{\mu} \\
& \boldsymbol{q} \geq 0.
\end{aligned} \tag{59}$$

If situation (ii) happens and one column of $A$ can be expressed as the linear combination of the other columns in $A$, then we denote the index of this column by $i'$ and we know that we can simply restrict $q_{i'} = 0$ without changing the objective value of (59). This is because $q_{i'}$ cannot be a basic variable otherwise the corresponding optimal basis will be linearly dependent. However, the index $i' \in \mathcal{I}$. This means that we cannot further delete $i'$ from the index set $\mathcal{I}$ by restricting $q_{i'} = 0$ without changing the objective value of LP (4). The above argument leads to a contradiction. Therefore, we know that situation (ii) cannot happen.

Given the arguments above, we know that the matrix $A$ is a non-singular matrix and thus the conditions in Lemma 2.1 are satisfied with the variable index set $\mathcal{I}$ and the constraint index set $\mathcal{J}$.

It only remains to show that we can tell whether the objective value of $V^{\mathrm{Infi}}$ or $\mathrm{Dual}_{\mathcal{I}}$ has changed by restricting one variable to be 0. Following the same steps as in Lemma 3.2, we can show that for index sets $\mathcal{I}'$ and $\mathcal{J}'$, it holds that

$$\mathrm{Gap}_1(N_0, \varepsilon) \geq V_{\mathcal{J}',\mathcal{I}'} - \bar{V}_{\mathcal{J}',\mathcal{I}',N_0} \tag{60}$$

and the upper gap as

$$\mathrm{Gap}_2(N_0, \varepsilon) \geq \bar{V}_{\mathcal{J}',\mathcal{I}',N_0} - V_{\mathcal{J}',\mathcal{I}'} \tag{61}$$

with the formulation of $\mathrm{Gap}_1(N_0, \varepsilon)$ and $\mathrm{Gap}_2(N_0, \varepsilon)$ given in (12) and (13). Therefore, as long as

$$2\mathrm{Gap}_1(N_0, \varepsilon) + 2\mathrm{Gap}_2(N_0, \varepsilon) \geq \min\{\Delta_1, \Delta_2\}, \tag{62}$$

where $\delta_1$ is defined in (16) and $\delta_2$ is defined in (19), we can tell whether $V^{\mathrm{Infi}}$ is different from $V_{\mathcal{I}'}^{\mathrm{Infi}}$ and whether $\mathrm{Dual}_{\mathcal{I}'}$ is different from $\mathrm{Dual}_{\mathcal{J}',\mathcal{I}'}$. Our proof is thus completed.

**Proof of Claim D.1.**    We first show that

$$\begin{aligned}
V = V_{\mathcal{I}} = \max \quad & (\hat{\boldsymbol{r}}_{\mathcal{I}})^\top \boldsymbol{q} \\
\text{s.t.} \quad & C(:, \mathcal{I})\boldsymbol{q} \leq \boldsymbol{\alpha} \\
& B(:, \mathcal{I})\boldsymbol{q} = \boldsymbol{\mu} \\
& \boldsymbol{q} \geq 0.
\end{aligned} \tag{63}$$

Denote by $\boldsymbol{q}^*$ the optimal solution to $V$ corresponding to the optimal basis $\mathcal{I}$. It is clear to see that $\boldsymbol{q}_{\mathcal{I}}^*$ is a feasible solution to $V_{\mathcal{I}}$ with the same objective value. Then, we have

$$V \leq V_{\mathcal{I}}. \tag{64}$$

On the other hand, we denote by $\hat{\boldsymbol{q}}$ one optimal solution to $V_{\mathcal{I}}$, and we construct

$$\tilde{\boldsymbol{q}}_{\mathcal{I}} = \hat{\boldsymbol{q}} \quad \text{and} \quad \tilde{\boldsymbol{q}}_{\mathcal{I}^c} = 0.$$

It is clear to see that $\tilde{\boldsymbol{q}}$ is a feasible solution to $V$ with the same objective value, which implies that

$$V_{\mathcal{I}} \leq V. \tag{65}$$

Therefore, (63) is proved from combining (64) and (65). The dual of $V_{\mathcal{I}}$ is given by

$$\begin{aligned}
\mathrm{Dual}_{\mathcal{I}} = \min \quad & \boldsymbol{\alpha}^\top \boldsymbol{y} + \boldsymbol{\mu}^\top \boldsymbol{z} \\
\text{s.t.} \quad & (C(:, \mathcal{I}))^\top \boldsymbol{y} + (B(:, \mathcal{I}))^\top \boldsymbol{z} \geq \hat{\boldsymbol{r}}(\mathcal{I}) \\
& \boldsymbol{y} \geq 0, \boldsymbol{z} \geq -\infty.
\end{aligned} \tag{66}$$

We denote by $\boldsymbol{y}^*, \boldsymbol{z}^*$ the optimal dual variable to $V$ corresponding to $\boldsymbol{q}^*$. It is easy to see that $\boldsymbol{y}^*, \boldsymbol{z}^*$ is also feasible to $\mathrm{Dual}_{\mathcal{I}}$ and

$$\mathrm{Dual}_{\mathcal{I}} = V_{\mathcal{I}} = V = \boldsymbol{\alpha}^\top \boldsymbol{y}^* + \boldsymbol{\mu}^\top \boldsymbol{z}^*$$

where the first equality follows from the strong duality between $V_{\mathcal{I}}$ and $\mathrm{Dual}_{\mathcal{I}}$, the second equality follows from (63), and the third equality follows from the strong duality between $V$ and its dual. Therefore, we know that $\boldsymbol{y}^*$ and $\boldsymbol{z}^*$ is also an optimal solution to $\mathrm{Dual}_{\mathcal{I}}$. Moreover, note that from the complementary slackness condition, since $\boldsymbol{q}_{\mathcal{I}}^* > 0$, we must have

$$(C(:, \mathcal{I}))^\top \boldsymbol{y}^* + (B(:, \mathcal{I}))^\top \boldsymbol{z}^* = \hat{\boldsymbol{r}}(\mathcal{I}). \tag{67}$$

We know that $\boldsymbol{y}^*, \boldsymbol{z}^*$ is a feasible solution to

$$\begin{aligned}
\mathrm{Dual}'_{\mathcal{I}} = \min \quad & \boldsymbol{\alpha}^\top \boldsymbol{y} + (1 - \gamma) \cdot \boldsymbol{\mu}^\top \boldsymbol{z} \\
\text{s.t.} \quad & (C(:, \mathcal{I}))^\top \boldsymbol{y} + (B(:, \mathcal{I}))^\top \boldsymbol{z} = \hat{\boldsymbol{r}}(\mathcal{I}) \\
& \boldsymbol{y} \geq 0, \boldsymbol{z} \geq -\infty,
\end{aligned} \tag{68}$$

which implies that

$$\mathrm{Dual}'_{\mathcal{I}} \leq \mathrm{Dual}_{\mathcal{I}}.$$

On the other hand, any feasible solution to $\mathrm{Dual}'_{\mathcal{I}}$ must be a feasible solution to $\mathrm{Dual}_{\mathcal{I}}$, and we have

$$\mathrm{Dual}'_{\mathcal{I}} \geq \mathrm{Dual}_{\mathcal{I}}.$$

Therefore, we must have $\mathrm{Dual}'_{\mathcal{I}} = \mathrm{Dual}_{\mathcal{I}}$ and our proof is completed. $\qquad\square$

# E  Proof of Theorem 5.2

We first prove the following lemma.

**Lemma E.1** *For the optimal basis identified in Algorithm 1 and the corresponding optimal solution $\boldsymbol{q}^*$, we denote by $\mathcal{I}^*$ and $\mathcal{J}^*$ the output sets as long as $N_0$ satisfies the condition (25). We also denote by $(\boldsymbol{y}^*, \boldsymbol{c}^*)$ the corresponding optimal dual solution. Then, it holds that*

$$N \cdot V_r(\pi^*, \mu_1) - \sum_{n=1}^{N} \hat{\boldsymbol{r}}^\top \mathbb{E}[\boldsymbol{q}^n] \leq \sum_{j \in \mathcal{J}^*} y_j^* \cdot \mathbb{E}[\alpha_j^N] + \sum_{s \in \mathcal{S}} z_s^* \cdot \mathbb{E}[\mu_s^N]. \tag{69}$$

Therefore, it suffices to analyze how the "remaining resources" $(\boldsymbol{\alpha}_{\mathcal{J}^*}^n, \boldsymbol{\mu}^n)$ behave. We now define

$$\tilde{\alpha}_k(n) = \frac{\alpha_k^n}{N-n}, \; \forall k \in \mathcal{J}^* \text{ and } \tilde{\mu}_s(n) = \frac{\mu_s^n}{N-n}, \; \forall s \in \mathcal{S}, \forall n \in [N]. \tag{70}$$

The key is to show that the stochastic process $\tilde{\alpha}_k(n)$ and $\tilde{\mu}_s$ possess some concentration properties such that they will stay within a small neighborhood of their initial value $\alpha_k^1$ and $\mu_s^1$ for a sufficiently long time. We denote by $\tau$ the time that one of $\tilde{\alpha}_k(n)$ for each $k \in \mathcal{J}^*$ and $\tilde{\mu}_s$ for each $s \in \mathcal{S}$ escape this neighborhood. Then, both $\text{Regret}_r(\pi, N)$ and $\text{Regret}_k(\pi, N)$ for each $k \in [K]$ can be upper bounded by $\mathbb{E}[N - \tau]$. From the update rule (22) and (23), we know that

$$\tilde{\alpha}_k(n+1) = \tilde{\alpha}_k(n) - \frac{\sum_{(s,a) \in \mathcal{I}^*} \boldsymbol{c}^n(s,a) \cdot q^n(s,a) - \tilde{\alpha}_k(n)}{N-n-1}, \; \forall k \in \mathcal{J}^* \tag{71}$$

and

$$\tilde{\mu}_s(n+1) = \tilde{\mu}_s(n) - \frac{\sum_{(s,a) \in \mathcal{I}^*} q^n(s,a) \cdot (\delta_{s',s} - \gamma \mathbb{1}_{\{s'=s^n(s,a)\}}) - \tilde{\mu}_s(n)}{N-n-1}. \tag{72}$$

Ideally, both $\tilde{\alpha}_k(n+1)$ and $\tilde{\mu}_s(n+1)$ will have the same expectation as $\tilde{\alpha}_k(n)$ and $\tilde{\mu}_s(n)$ such that they become a martingale. However, this is not true since we have estimation error over $C(\mathcal{J}^*, \mathcal{I}^*)$ and $B(:, \mathcal{I}^*)$, and we only use their estimates to compute $\boldsymbol{q}^n$. Nevertheless, we can show that $\tilde{\alpha}_k(n)$ for each $k \in \mathcal{J}^*$ and $\tilde{\mu}_s$ for each $s \in \mathcal{S}$ behave as a sub-martingale. Then, from the concentration property of the sub-martingale, we upper bound $\mathbb{E}[\alpha_k^N]$ for each $k \in \mathcal{J}^*$ and $\mathbb{E}[\mu_s^N]$ for each $s \in \mathcal{S}$. The term $|\mathbb{E}[\alpha_k^N]|$ for each $k \in [K] \backslash \mathcal{J}^*$ can be upper bounded as well. The results are presented in the following lemma.

**Lemma E.2** *Denote by $\bar{\pi}^N$ the output policy of Algorithm 2 and denote by $N$ the number of rounds. Then, it holds that*

$$N \cdot V_r(\pi^*, \mu_1) - \sum_{n=1}^{N} \hat{\boldsymbol{r}}^\top \mathbb{E}[\boldsymbol{q}^n] \leq O\left(\frac{(|\mathcal{S}| + K)^3}{\alpha \cdot \sigma \cdot \min\{\sigma^2, (1-\gamma)^2 \cdot \Delta\}} \cdot \frac{\log(N)}{N}\right)$$

*where the parameters $\alpha = \min_{k \in [K]}\{\alpha_k\}$, $\Delta = \min\{\Delta_1^2, \Delta_2^2\}$ with $\Delta_1$ given in (16) and $\Delta_2$ given in (19), $\sigma$ given in (27). Also, for any $k \in [K]$, we have*

$$N \cdot \alpha_k - \sum_{n=1}^{N} \hat{\boldsymbol{c}}_k^\top \mathbb{E}[\boldsymbol{q}^n] \leq O\left(\frac{(|\mathcal{S}| + K)^3}{\alpha \cdot \sigma \cdot \min\{\sigma^2, (1-\gamma)^2 \cdot \Delta\}} \cdot \frac{\log(N)}{N}\right). \tag{73}$$

## E.1  Proof of Lemma E.1

Note that the distribution of $\boldsymbol{q}^n$ is independent of the distribution of $\boldsymbol{r}^n$. We know that

$$\mathbb{E}\left[\sum_{n=1}^{N} (\boldsymbol{r}^n)^\top \boldsymbol{q}^n\right] = \sum_{n=1}^{N} (\hat{\boldsymbol{r}})^\top \mathbb{E}[\boldsymbol{q}^n] = \sum_{n=1}^{N} (\hat{\boldsymbol{r}}_{\mathcal{I}^*})^\top \mathbb{E}[\boldsymbol{q}_{\mathcal{I}^*}^n]$$

Denote by $\boldsymbol{q}^*$ and $\boldsymbol{y}^*, \boldsymbol{z}^*$ the optimal primal-dual variable corresponding to the optimal basis $\mathcal{I}^*$ and $\mathcal{J}^*$. From the complementary slackness condition and noting that $\boldsymbol{q}_{\mathcal{I}^*}^* > 0$, we know that

$$(C(\mathcal{J}^*, \mathcal{I}^*))^\top \boldsymbol{y}_{\mathcal{J}^*}^* + (B(:, \mathcal{I}^*))^\top \boldsymbol{z}^* = \hat{\boldsymbol{r}}_{\mathcal{I}^*}. \tag{74}$$

Also, we can define a matrix $C^n(\mathcal{J}^*, \mathcal{I}^*)$ such that the element of $C^n(\mathcal{J}^*, \mathcal{I}^*)$ at the $k \in \mathcal{J}^*$ row and $(s,a) \in \mathcal{I}^*$ column is $c_k^n(s,a)$. We can also define a matrix $B^n(:, \mathcal{I}^*)$ such that the element of $B^n(:, \mathcal{I}^*)$ at the $s' \in \mathcal{S}$ row and $(s,a) \in \mathcal{I}^*$ column is $\delta_{s',s} - \gamma \cdot \mathbb{1}_{\{s' = s^n(s,a)\}}$. It is easy to see that

$$\mathbb{E}[C^n(\mathcal{J}^*, \mathcal{I}^*)] = C(\mathcal{J}^*, \mathcal{I}^*) \text{ and } \mathbb{E}[B^n(:, \mathcal{I}^*)] = B(:, \mathcal{I}^*).$$

Then, it holds that

$$
\begin{aligned}
\sum_{n=1}^{N} (\hat{\boldsymbol{r}}_{\mathcal{I}^*})^\top \mathbb{E}\left[\boldsymbol{q}_{\mathcal{I}^*}^n\right] &= \sum_{n=1}^{N} \left((C(\mathcal{J}^*, \mathcal{I}^*))^\top \boldsymbol{y}_{\mathcal{J}^*}^* + (B(:, \mathcal{I}^*))^\top \boldsymbol{z}^*\right)^\top \mathbb{E}[\boldsymbol{q}_{\mathcal{I}^*}^n] \\
&= \mathbb{E}\left[\sum_{n=1}^{N} \left((C^n(\mathcal{J}^*, \mathcal{I}^*))^\top \boldsymbol{y}_{\mathcal{J}^*}^* + (B^n(:, \mathcal{I}^*))^\top \boldsymbol{z}^*\right)^\top \boldsymbol{q}_{\mathcal{I}^*}^n\right] \\
&= \mathbb{E}\left[\sum_{n=1}^{N} \left((\boldsymbol{y}^*)^\top (C^n(\mathcal{J}^*, \mathcal{I}^*))\boldsymbol{q}_{\mathcal{I}^*}^n + (\boldsymbol{z}^*)^\top (B^n(:, \mathcal{I}^*))\boldsymbol{q}_{\mathcal{I}^*}^n\right)\right]
\end{aligned}
\tag{75}
$$

Note that we have

$$\sum_{n=1}^{N} C^n(\mathcal{J}^*, \mathcal{I}^*))\boldsymbol{q}_{\mathcal{I}^*}^n = \boldsymbol{\alpha}_{\mathcal{J}^*}^1 - \boldsymbol{\alpha}^N \tag{76}$$

and

$$\sum_{n=1}^{N} B^n(:, \mathcal{I}^*))\boldsymbol{q}_{\mathcal{I}^*}^n = \boldsymbol{\mu}^1 - \boldsymbol{\mu}^N. \tag{77}$$

Plugging (76) and (77) back into (75), we get that

$$\sum_{n=1}^{N} (\hat{\boldsymbol{r}}_{\mathcal{I}^*})^\top \mathbb{E}\left[\boldsymbol{q}_{\mathcal{I}^*}^n\right] = (\boldsymbol{y}_{\mathcal{J}^*}^*)^\top \boldsymbol{\alpha}_{\mathcal{J}^*}^1 + (\boldsymbol{z}^*)^\top \boldsymbol{\mu}^1 - (\boldsymbol{y}_{\mathcal{J}^*}^*)^\top \mathbb{E}\left[\boldsymbol{\alpha}_{\mathcal{J}^*}^N\right] - (\boldsymbol{z}^*)\top\mathbb{E}\left[\boldsymbol{\mu}^N\right].$$

Note that

$$V_r(\pi^*, \mu_1) = (\boldsymbol{y}_{\mathcal{J}^*}^*)^\top \boldsymbol{\alpha}_{\mathcal{J}^*}^1 + (\boldsymbol{z}^*)^\top \boldsymbol{\mu}^1$$

that holds from the strong duality of $V^{\text{Infi}}$ (4). Then, we have that

$$N \cdot V_r(\pi^*, \mu_1) - \sum_{n=1}^{N} (\hat{\boldsymbol{r}})^\top \mathbb{E}[\boldsymbol{q}^n] \leq (\boldsymbol{y}_{\mathcal{J}^*}^*)^\top \mathbb{E}\left[\boldsymbol{\alpha}_{\mathcal{J}^*}^N\right] + (\boldsymbol{z}^*)^\top \mathbb{E}\left[\boldsymbol{\mu}^N\right]. \tag{78}$$

Our proof is thus completed.

### E.2 Proof of Lemma E.2

We now condition on the event that $|\bar{r}_n(s,a) - \hat{r}(s,a)|$, $|\bar{c}_{k,n}(s,a) - \hat{c}_k(s,a)|$, and $|\bar{P}_n(s'|s,a) - P(s'|s,a)|$ for each $k \in [K]$, $(s,a) \in \mathcal{S} \times \mathcal{A}$ and $s' \in \mathcal{S}$ are all bounded by $\mathrm{Rad}(n, \varepsilon)$, for any $n \in [N]$. We know that this event happens with probability at least $1 - N \cdot (K|\mathcal{S}||\mathcal{A}| - |\mathcal{S}|^2|\mathcal{A}|) \cdot \varepsilon$. From now on, we set $\varepsilon = \frac{1}{N^2}$ to guarantee that the event happens with probability at least $1 - \frac{K|\mathcal{S}||\mathcal{A}| + |\mathcal{S}|^2|\mathcal{A}|}{N}$.

We consider the stochastic process $\tilde{\alpha}_k(n)$ and $\tilde{\mu}_s(n)$ defined in (70). For a fixed $\nu > 0$ which we specify later, we define a set

$$\mathcal{X} = \{\boldsymbol{\alpha}' \in \mathbb{R}^{|\mathcal{J}^*|} : \alpha_k' \in [\alpha_k - \nu, \alpha_k + \nu], \forall k \in \mathcal{J}^*\}, \tag{79}$$

and

$$\mathcal{Y} = \{\boldsymbol{\mu}' \in \mathbb{R}^{|\mathcal{S}|} : \mu_s' \in [\mu_s - \nu, \mu_s + \nu], \forall s \in \mathcal{S}\}. \tag{80}$$

It is easy to see that initially, $\tilde{\boldsymbol{\alpha}}(1) \in \mathcal{X}$ and $\tilde{\boldsymbol{\mu}}(1) \in \mathcal{Y}$. We show that $\tilde{\boldsymbol{\alpha}}(n)$ and $\tilde{\boldsymbol{\mu}}(n)$ behave well as long as they stay in the region $\mathcal{X}$ and $\mathcal{Y}$ for a sufficiently long time. To this end, we define a stopping time

$$\tau = \min_{n \in [N]} \{\tilde{\boldsymbol{\alpha}}(n) \notin \mathcal{X} \text{ or } \tilde{\boldsymbol{\mu}}(n) \notin \mathcal{Y}\}. \tag{81}$$

Note that in Algorithm 2, to stop $\boldsymbol{q}^n$ from behaving ill when $n$ is small, we project it to a set that guarantees $\|\boldsymbol{q}^n\|_1 \leq 2$. We now show in the following claim that when $n$ is large enough but smaller than the stopping time $\tau$, there is no need to do projection.

**Claim E.3** *There exist two constants $N_0'$ and $\nu_0$. When $\max\{N_0, N_0'\} \le n \le \tau$, and $\nu \le \nu_0$, it holds that $\|\tilde{q}_{\mathcal{I}^*}^n\|_1 \le 2$, where $\tilde{q}_{\mathcal{I}^*}^n$ denotes the solution to the linear equations (21). Specifically, $N_0$ is given in (25) and $N_0'$ is given as follows*

$$N_0' = 16 \cdot \frac{(|\mathcal{S}| + K)^2}{\sigma^2} \cdot \log(1/\varepsilon) \tag{82}$$

*Also, $\nu_0$ is given as follows*

$$\nu_0 := 16 \cdot \frac{\sigma \cdot (1 - \gamma)}{(|\mathcal{S}| + K)^2}. \tag{83}$$

We set $\nu$ satisfy the condition $\nu \le \nu_0$ with $\nu_0$ satisfies the condition in Claim E.3. We bound $\mathbb{E}[N - \tau]$ in the following claim.

**Claim E.4** *Let the stopping time $\tau$ be defined in (81). It holds that*

$$\mathbb{E}[N - \tau] \le \max\{N_0, N_0'\} + 2(K + |\mathcal{S}|) \cdot \exp(-\nu^2/8)$$

*where $N_0$ is given in (25) and $N_0'$ is given in (82), as long as*

$$N \ge \max\{N_0, N_0'\} \ \text{ and } \ N \ge \frac{8}{\nu^2} \ge \frac{8}{\nu_0^2} = \frac{(|\mathcal{S}| + K)^4}{32\sigma^2 \cdot (1 - \gamma)^2}. \tag{84}$$

*Also, for any $N'$ such that $\max\{N_0, N_0'\} \le N' \le N$, it holds that*

$$P(\tau \le N') \le \frac{(K + |\mathcal{S}|) \cdot \nu^2}{4} \cdot \exp\left(-\frac{\nu^2 \cdot (N - N' + 1)}{8}\right). \tag{85}$$

From the definition of the stopping time $\tau$ in (81), we know that for each $k \in \mathcal{J}^*$, it holds

$$\alpha_k^{\tau-1} \in [(N - \tau + 1) \cdot (\alpha_k - \nu), (N - \tau + 1) \cdot (\alpha_k + \nu)]$$

Thus, we have that

$$|\alpha_k^N| \le |\alpha_k^{\tau-1}| + \sum_{t=\tau}^{N} \sum_{(s,a)\in\mathcal{I}^*} c_k^n(s,a) \cdot q^n(s,a) \tag{86}$$

and thus

$$\left|\mathbb{E}[\alpha_k^N]\right| \le 4\mathbb{E}[N - \tau] \le 4\max\{N_0, N_0'\} + 8(K + |\mathcal{S}|) \cdot \exp(-\nu^2/8). \tag{87}$$

Following the same procedure, we can show that for each $s \in \mathcal{S}$, it holds that

$$\left|\mathbb{E}[\mu^N(s)]\right| \le 4\mathbb{E}[N - \tau] \le 4\max\{N_0, N_0'\} + 8(K + |\mathcal{S}|) \cdot \exp(-\nu^2/8). \tag{88}$$

We finally consider the other constraints $k \in \mathcal{J}^{*c}$. Note that following the definition of $\boldsymbol{\alpha}^n$ and $\boldsymbol{\mu}^n$, we have that

$$A^* \cdot \left(\sum_{n=1}^{N} \mathbb{E}[q_{\mathcal{I}^*}^n]\right) = [\boldsymbol{\alpha}_{\mathcal{J}^*}^1 - \mathbb{E}\left[\boldsymbol{\alpha}_{\mathcal{J}^*}^N\right]; \boldsymbol{\mu}^1 - \mathbb{E}\left[\boldsymbol{\mu}^N\right]]. \tag{89}$$

Also, from the bindingness of $\boldsymbol{q}^*$ regarding the optimal basis $\mathcal{I}^*$ and $\mathcal{J}^*$, we have

$$N \cdot A^* \cdot \boldsymbol{q}_{\mathcal{I}^*}^* = [\boldsymbol{\alpha}_{\mathcal{J}^*}^1; \boldsymbol{\mu}^1]. \tag{90}$$

Therefore, it holds that

$$\sum_{n=1}^{N} \mathbb{E}\left[q_{\mathcal{I}^*}^n\right] = N \cdot \boldsymbol{q}_{\mathcal{I}^*}^* - (A^*)^{-1} \cdot \left[\mathbb{E}\left[\boldsymbol{\alpha}_{\mathcal{J}^*}^N\right]; \mathbb{E}\left[\boldsymbol{\mu}^N\right]\right], \tag{91}$$

and

$$\left\|\sum_{n=1}^{N} \mathbb{E}\left[q_{\mathcal{I}^{*c}}^n\right]\right\|_1 \le 2N_0. \tag{92}$$

Finally, for any $k \in \mathcal{J}^{*c}$, we have

$$(\hat{c}_k)^\top \left(\sum_{n=1}^{N} \mathbb{E}\left[q^n\right]\right) = N \cdot (\hat{c}_k)^\top \boldsymbol{q}_{\mathcal{I}^*}^* - (\hat{c}_k)^\top \cdot (A^*)^{-1} \cdot \left[\mathbb{E}\left[\boldsymbol{\alpha}_{\mathcal{J}^*}^N\right]; \mathbb{E}\left[\boldsymbol{\mu}^N\right]\right] + 2N_0$$

$$= N \cdot (\hat{c}_k)^\top \boldsymbol{q}^* - (\hat{c}_k)^\top \cdot (A^*)^{-1} \cdot \left[\mathbb{E}\left[\boldsymbol{\alpha}_{\mathcal{J}^*}^N\right]; \mathbb{E}\left[\boldsymbol{\mu}^N\right]\right] + 2N_0 \tag{93}$$

From the feasibility of $\boldsymbol{q}^*$, we know that

$$N \cdot \alpha_k \geq N \cdot (\hat{\boldsymbol{c}}_k)^\top \boldsymbol{q}^*.$$

Therefore, for any $k \in \mathcal{J}^{*c}$, it holds that

$$N \cdot \alpha_k - \sum_{n=1}^{N} \hat{\boldsymbol{c}}_k^\top \mathbb{E}\left[\boldsymbol{q}^n\right] \leq (\hat{\boldsymbol{c}}_k)^\top \cdot (A^*)^{-1} \cdot \left[\mathbb{E}[\boldsymbol{\alpha}_{\mathcal{J}^*}^N]; \mathbb{E}[\boldsymbol{\mu}^N]\right] + 2N_0$$

$$\leq \frac{K + |\mathcal{S}|}{\sigma} \cdot \left(4\max\{N_0, N_0'\} + 8(K + |\mathcal{S}|) \cdot \exp(-\nu^2/8)\right) + 2N_0. \tag{94}$$

Moreover, the definition of $\sigma$ in (27) implies that following upper bound on the norm of the dual variable $\boldsymbol{y}^*$ and $\boldsymbol{z}^*$.

$$\|\boldsymbol{y}^*\|_1 + \|\boldsymbol{z}^*\|_1 = \|(A^{*\top})^{-1} \cdot \hat{\boldsymbol{r}}_{\mathcal{J}^*}\|_1 \leq \frac{|\mathcal{S}| + K}{\sigma}. \tag{95}$$

Therefore, we know that the regret over the reward and the regret over the constraint violation can all be bounded by using (87), (88), and (94). We present the bounds as follows.

$$N \cdot V_r(\pi^*, \mu_1) - \sum_{n=1}^{N} \hat{\boldsymbol{r}}^\top \mathbb{E}[\boldsymbol{q}^n] \leq \frac{K + |\mathcal{S}|}{\sigma} \cdot \left(4\max\{N_0, N_0'\} + 8(K + |\mathcal{S}|) \cdot \exp(-\nu^2/8)\right). \tag{96}$$

Meanwhile, the constraint violations are bounded by (87), (88), and (94). Our proof is thus completed.

**Proof of Claim E.3.** Denote by $\boldsymbol{q}^*$ the optimal solution corresponding to the optimal basis $\mathcal{I}^*$ and $\mathcal{J}^*$. Then, it holds that

$$\begin{bmatrix} C(\mathcal{J}^*, \mathcal{I}^*) \\ B(:, \mathcal{I}^*) \end{bmatrix} \cdot \boldsymbol{q}_{\mathcal{I}^*}^* = \begin{bmatrix} \boldsymbol{\alpha}_{\mathcal{J}^*} \\ \boldsymbol{\mu} \end{bmatrix}. \tag{97}$$

We compare $\tilde{\boldsymbol{q}}_{\mathcal{I}^*}^n$ with $\boldsymbol{q}_{\mathcal{I}^*}^*$ when $n$ large enough. Note that when $n \geq N_0$, $\tilde{\boldsymbol{q}}^n$ is the solution to the following linear equations

$$\begin{bmatrix} \bar{C}^n(\mathcal{J}^*, \mathcal{I}^*) \\ \bar{B}^n(:, \mathcal{I}^*) \end{bmatrix} \cdot \tilde{\boldsymbol{q}}_{\mathcal{I}^*}^n = \begin{bmatrix} \dfrac{\boldsymbol{\alpha}_{\mathcal{J}^*}^n}{N - n + 1} \\ \dfrac{\boldsymbol{\mu}^n}{N - n + 1} \end{bmatrix}. \tag{98}$$

When $n \leq \tau$, we know that

$$\left| \boldsymbol{\alpha}_{\mathcal{J}^*} - \frac{\boldsymbol{\alpha}_{\mathcal{J}^*}^n}{N - n + 1} \right| \leq \nu \tag{99}$$

and

$$\left| \boldsymbol{\mu} - \frac{\boldsymbol{\mu}^n}{N - n + 1} \right| \leq \nu. \tag{100}$$

Moreover, we know that the absolute value of each element of $\bar{C}^n(\mathcal{J}^*, \mathcal{I}^*) - C(\mathcal{J}^*, \mathcal{I}^*)$, and $\bar{B}^n(:, \mathcal{I}^*) - B(:, \mathcal{I}^*)$ is upper bounded by $\mathrm{Rad}(n, \varepsilon)$. We now bound the distance between the solutions to the linear equations (97) and (98). The perturbation of the matrix is denoted as

$$\Delta A^* = \begin{bmatrix} C(\mathcal{J}^*, \mathcal{I}^*) - \bar{C}^n(\mathcal{J}^*, \mathcal{I}^*) \\ B(:, \mathcal{I}^*) - \bar{B}(:, \mathcal{I}^*) \end{bmatrix}.$$

Clearly, it holds that

$$\|\Delta A^*\|_1 \leq \mathrm{Rad}(n, \varepsilon) \cdot (K + |\mathcal{S}|). \tag{101}$$

Therefore, as long as

$$\|\Delta A^*\|_1 \leq \mathrm{Rad}(n, \varepsilon) \cdot (K + |\mathcal{S}|) \leq \frac{1}{2\|(A^*)^{-1}\|_1} \leq \frac{1}{2\sigma}, \tag{102}$$

following standard perturbation analysis of linear equations [28], we have that

$$\frac{\|\tilde{\boldsymbol{q}}_{\mathcal{I}^*}^n - \boldsymbol{q}_{\mathcal{I}^*}^*\|_1}{\|\boldsymbol{q}_{\mathcal{I}^*}^*\|_1} \leq \frac{\kappa(A^*)}{1 - \kappa(A^*) \cdot \frac{\|\Delta A^*\|_1}{\|A^*\|_1}} \cdot \left( \frac{\|\Delta A^*\|_1}{\|A^*\|_1} + \frac{(|\mathcal{S} + K| \cdot \nu)}{\|[\boldsymbol{\alpha}_{\mathcal{J}^*}; \boldsymbol{\mu}]\|_1} \right)$$

$$\leq 2 \cdot \kappa(A^*) \cdot \left( \frac{\|\Delta A^*\|_1}{\|A^*\|_1} + \frac{(|\mathcal{S} + K| \cdot \nu)}{\|[\boldsymbol{\alpha}_{\mathcal{J}^*}; \boldsymbol{\mu}]\|_1} \right) \tag{103}$$

$$\leq 2 \cdot \kappa(A^*) \cdot \left( \frac{\|\Delta A^*\|_1}{\|A^*\|_1} + \frac{(|\mathcal{S}| + K)) \cdot \nu}{1 - \gamma} \right),$$

where $\kappa(A^*) = \|A^*\|_1 \cdot \|(A^*)^{-1}\|_1$ denotes the conditional number of $A^*$. The last inequality follows from $\|[\boldsymbol{\alpha}_{\mathcal{J}^*}; \boldsymbol{\mu}]\|_1 \geq 1 - \gamma$. Further, note that $\|\boldsymbol{q}_{\mathcal{I}^*}^*\|_1 = 1$. Therefore, in order to satisfy the condition $\|\tilde{\boldsymbol{q}}_{\mathcal{I}^*}^n\|_1 \leq 2$, we only need the right hand side of (103) to be upper bounded by 1. Clearly, as long as $n$ satisfies the condition (102) and the following condition

$$2 \cdot \kappa(A^*) \cdot \frac{\|\Delta A^*\|_1}{\|A\|_1} \leq 2 \cdot \frac{\text{Rad}(n, \varepsilon) \cdot (K + |\mathcal{S}|)}{\sigma} \leq \frac{1}{2}, \tag{104}$$

we only need to select a $\nu$ such that

$$2 \cdot \kappa(A^*) \cdot \frac{(|\mathcal{S}| + K) \cdot \nu}{1 - \gamma} \leq \frac{1}{2}. \tag{105}$$

Combining (102) and (104), we know that $n$ needs to satisfy the following conditions: $n \geq N_0$ and

$$n \geq N_0' := 16 \cdot \frac{(|\mathcal{S}| + K)^2}{\sigma^2} \cdot \log(1/\varepsilon). \tag{106}$$

Also, $\nu$ is selected to satisfy the following condition

$$\nu \leq \nu_0 := 16 \cdot \frac{\sigma \cdot (1 - \gamma)}{(|\mathcal{S}| + K)^2}. \tag{107}$$

Our proof is thus completed. $\qquad\square$

**Proof of Claim E.4.** Now we fix a $k \in \mathcal{J}^*$. We specify a $\bar{N}_0 = \max\{N_0, N_0'\}$. For any $\bar{N}_0 \leq N' \leq N$, it holds that

$$\tilde{\alpha}_k(N') - \tilde{\alpha}_k(\bar{N}_0) = \sum_{n=\bar{N}_0}^{N'-1} (\tilde{\alpha}_k(n+1) - \tilde{\alpha}_k(n)).$$

We define $\xi_k(n) = \tilde{\alpha}_k(n+1) - \tilde{\alpha}_k(n)$. Then, we have

$$\tilde{\alpha}_k(N') - \tilde{\alpha}_k(\bar{N}_0) = \sum_{n=\bar{N}_0}^{N'-1} (\xi_k(n) - \mathbb{E}[\xi_k(n)|\mathcal{F}_n]) + \sum_{n=\bar{N}_0}^{N'-1} \mathbb{E}[\xi_k(n)|\mathcal{F}_n].$$

where $\mathcal{F}_n$ denotes the filtration of information up to step $n$. Note that due to the update in (71), we have

$$\xi_k(n) = \frac{\tilde{\alpha}_k(n) - \sum_{(s,a) \in \mathcal{I}^*} c_k^n(s,a) \cdot q^n(s,a)}{N - n - 1}.$$

Then, it holds that

$$|\xi_k(n) - \mathbb{E}[\xi_k(n)|\mathcal{F}_n]| \leq \frac{2}{N - n + 1} \tag{108}$$

where the inequality follows from the fact that the value of $\tilde{\alpha}_k(n)$ is deterministic given the filtration $\mathcal{F}_n$ and $\|q^n\|_1 \leq 2$ for any $n$. Note that

$$\{\xi_k(n) - \mathbb{E}[\xi_k(n)|\mathcal{F}_n]\}_{\forall n = \bar{N}_0, \dots, N'}$$

forms a martingale difference sequence. Following Hoeffding's inequality, for any $N'' \leq N'$ and any $b > 0$, it holds that

$$P\left( \left| \sum_{n=\bar{N}_0}^{N''} (\xi_k(n) - \mathbb{E}[\xi_k(n)|\mathcal{F}_n]) \right| \geq b \right) \leq 2 \exp\left( -\frac{b^2}{2 \cdot \sum_{n=\bar{N}_0}^{N''} 1/(N - n + 1)^2} \right)$$

$$\leq 2 \exp\left( -\frac{b^2 \cdot (N - N'' + 1)}{2} \right).$$

Therefore, we have that

$$
P\left(\left|\sum_{n=\bar{N}_0}^{N''} (\xi_k(n) - \mathbb{E}[\xi_k(n)|\mathcal{F}_n])\right| \ge b \text{ for some } \bar{N}_0 \le N'' \le N'\right)
$$

$$
\le \sum_{N''=\bar{N}_0}^{N'} 2\exp\left(-\frac{b^2 \cdot (N - N'' + 1)}{2}\right) \le b^2 \cdot \exp\left(-\frac{b^2 \cdot (N - N' + 1)}{2}\right) \tag{109}
$$

holds for any $b > 0$.

We now bound the probability that $\tau > N'$ for one particular $N'$ such that $\bar{N}_0 \le N' \le N$. Suppose that $N' \le \tau$, then, from Claim E.3, for each $n \le N'$, we know that $\|\tilde{q}^n\|_1 \le 2$ and therefore $q^n = \tilde{q}^n$ as the solution to (21). We have

$$
\tilde{\alpha}_k(n) = \sum_{(s,a)\in\mathcal{I}^*} \bar{c}_{k,n}(s,a) \cdot q^n(s,a).
$$

It holds that

$$
|\mathbb{E}[\xi_k(n)|\mathcal{F}_n]| \le \frac{1}{N-n+1} \cdot \sum_{(s,a)\in\mathcal{I}^*} q^n(s,a) \cdot |\mathbb{E}[\bar{c}_{k,n}(s,a)] - \hat{c}_k^n(s,a)| \le \frac{2\mathrm{Rad}(n,\varepsilon)}{N-n+1}. \tag{110}
$$

Then, we know that

$$
\sum_{n=\bar{N}_0}^{N'-1} |\mathbb{E}[\xi_k(n)|\mathcal{F}_n]| \le \sqrt{\frac{\log(2/\varepsilon)}{2}} \cdot \sum_{n=\bar{N}_0}^{N'-1} \frac{1}{\sqrt{n}\cdot(N-n)} \le \sqrt{\frac{\log(2/\varepsilon)}{2}} \cdot \sqrt{N'-1} \cdot \sum_{n=\bar{N}_0}^{N'-1} \frac{1}{n\cdot(N-n)}
$$

$$
= \sqrt{\frac{\log(2/\varepsilon)}{2}} \cdot \frac{\sqrt{N'-1}}{N} \cdot \sum_{n=\bar{N}_0}^{N'-1} \left(\frac{1}{n} + \frac{1}{N-n}\right)
$$

$$
\le \sqrt{2\log(2/\varepsilon)} \cdot \frac{\sqrt{N'-1}}{N} \cdot \log(N) \le \frac{\sqrt{2\log(2/\varepsilon)}}{\sqrt{N}} \cdot \log(N) \le \frac{\nu}{2} \tag{111}
$$

for a $N$ large enough such that

$$
N \ge \frac{8}{\nu^2} \ge \frac{8}{\nu_0^2} = \frac{(|\mathcal{S}|+K)^4}{32\sigma^2 \cdot (1-\gamma)^2} \tag{112}
$$

Combining (111) and (109) with $b = \nu/2$, and apply a union bound over all $k \in \mathcal{J}^*$ and $s \in \mathcal{S}$, we know that

$$
P(\tau \le N') \le \frac{(K+|\mathcal{S}|)\cdot\nu^2}{4} \cdot \exp\left(-\frac{\nu^2 \cdot (N-N'+1)}{8}\right). \tag{113}
$$

Therefore, we know that

$$
\mathbb{E}[N-\tau] = \sum_{N'=1}^{N} P(\tau \le N') \le \bar{N}_0 + \sum_{N'=\bar{N}_0}^{N} P(\tau \le N') \le \bar{N}_0 + 2(K+|\mathcal{S}|)\cdot\exp(-\nu^2/8)
$$

which completes our proof. $\qquad\square$

### E.3   Final Proof of Theorem 5.2

Note that in the proof of Lemma E.2, we have shown the following bounds.

$$
\left|\mathbb{E}[\alpha_k^N]\right| \le 4\mathbb{E}[N-\tau] \le 4\max\{N_0, N_0'\} + 8(K+|\mathcal{S}|)\cdot\exp(-\nu^2/8). \tag{114}
$$

holds for the each $k \in \mathcal{J}^*$. For each $s \in \mathcal{S}$, it holds that

$$
\left|\mathbb{E}[\mu^N(s)]\right| \le 4\mathbb{E}[N-\tau] \le 4\max\{N_0, N_0'\} + 8(K+|\mathcal{S}|)\cdot\exp(-\nu^2/8). \tag{115}
$$

The caveat of directly transferring the bound of (114) and (115) into the sample complexity bounds of the policy $\bar{\pi}^N$ is that, the vector $\bar{q}^N$ does not directly characterize an occupancy measure. This point

can be seen by noting that there is a gap between $B\bar{q}^N$ and $\boldsymbol{\mu}$, though bounded by $O(\log(N)/N)$ by setting $\varepsilon = 1/N^2$. However, we can show that the gap between $\bar{q}^N$ and $q^*$ is upper bounded by $O(\log(N)/N)$, which implies a bound over the gap between the policy $\bar{\pi}^N$ and the optimal policy $\pi^*$ that corresponds to the occupancy measure $q^*$. This bound over the gap between the policy distributions can be then transferred into the bound over the gap between the state-value functions under the policy $\bar{\pi}^N$ and $\pi^*$. The regret bounds can be obtained then.

We first bound the gap between $\bar{q}^N$ and $q^*$. Note that as long as $n \geq N_0$, we have $\mathcal{I}_n = \mathcal{I}^*$ following Theorem 5.1. Then, by noting $\mathcal{I}_N = \mathcal{I}^*$, we know that

$$\bar{q}^N_{\mathcal{I}^{*c}} = q^*_{\mathcal{I}^{*c}}. \tag{116}$$

Also, note that following the definition of $\boldsymbol{\alpha}^n$ and $\boldsymbol{\mu}^n$, we have that

$$A^* \cdot \left( \sum_{n=1}^{N} \mathbb{E}[q^n_{\mathcal{I}^*}] \right) = [\boldsymbol{\alpha}^1_{\mathcal{J}^*} - \mathbb{E}\left[\boldsymbol{\alpha}^N_{\mathcal{J}^*}\right] ; \boldsymbol{\mu}^1 - \mathbb{E}\left[\boldsymbol{\mu}^N\right]]. \tag{117}$$

Also, from the bindingness of $q^*$ regarding the optimal basis $\mathcal{I}^*$ and $\mathcal{J}^*$, we have

$$N \cdot A^* \cdot q^*_{\mathcal{I}^*} = [\boldsymbol{\alpha}^1_{\mathcal{J}^*} ; \boldsymbol{\mu}^1]. \tag{118}$$

Then, from (117) and (118), we know that

$$\begin{aligned}
\left\| \mathbb{E}\left[\bar{q}^N_{\mathcal{I}^*}\right] - q^*_{\mathcal{I}^*} \right\|_\infty &= \left\| q^*_{\mathcal{I}^*} - \frac{1}{N} \cdot \sum_{n=1}^{N} \mathbb{E}[q^n_{\mathcal{I}^*}] \right\|_\infty = \frac{\left\| (A^*)^{-1} \cdot \left[\mathbb{E}\left[\boldsymbol{\alpha}^N_{\mathcal{J}^*}\right] ; \mathbb{E}\left[\boldsymbol{\mu}^N\right]\right] \right\|_\infty}{N} \\
&\leq \frac{\left\| \left[\mathbb{E}\left[\boldsymbol{\alpha}^N_{\mathcal{J}^*}\right] ; \mathbb{E}\left[\boldsymbol{\mu}^N\right]\right] \right\|_\infty}{\sigma \cdot N} \\
&\leq \frac{1}{\sigma \cdot N} \cdot \left( 4 \max\{N_0, N_0'\} + 8(K + |\mathcal{S}|) \cdot \exp(-\nu^2/8) \right).
\end{aligned} \tag{119}$$

From Markov's inequality, for each $i \in \mathcal{I}^*$ and any $a > 0$, we know that

$$P\left( |\bar{q}^N_i - q^*_i| > g \right) \leq \frac{1}{g \cdot \sigma \cdot N} \cdot \left( 4 \max\{N_0, N_0'\} + 8(K + |\mathcal{S}|) \cdot \exp(-\nu^2/8) \right). \tag{120}$$

We denote by

$$\xi = \min_{(s,a) \in \mathcal{I}^*} \{q^*(s,a)\}. \tag{121}$$

The policy $\bar{\pi}^N$ is essentially random by noting that $q^N$ is a random variable, where the randomness comes from the randomness of the filtration $\mathcal{F}_N$. For each $s \in \mathcal{S}$ and $a \in \mathcal{A}$, we denote by $\bar{\pi}(a|s)$ the (ex-ante) probability that the random policy $\bar{\pi}$ takes the action $a$ given the state $s$. Then, for any $0 < g \leq \xi/2$, we note that

$$|q^N_i - q^*_i| \leq g \text{ for each } i = (s,a) \in \mathcal{I}^* \text{ implies that } \left| \frac{q^N(s,a)}{\sum_{a' \in \mathcal{A}} q^N(s,a')} - \frac{q^*(s,a)}{\sum_{a' \in \mathcal{A}} q^*(s,a')} \right| \leq \frac{2g}{\xi}, \tag{122}$$

for each $i = (s,a) \in \mathcal{I}^*$. For any $0 < g \leq \xi/2$, note that

$$P\left( |q^N_i - q^*_i| \leq g \text{ for each } i = (s,a) \in \mathcal{I}^* \right) \geq 1 - \frac{|\mathcal{S}| + K}{g \cdot \sigma \cdot N} \cdot \left( 4 \max\{N_0, N_0'\} + 8(K + |\mathcal{S}|) \cdot \exp(-\nu^2/8) \right), \tag{123}$$

where the inequality follows from the bound (120) and the union bound over $i \in \mathcal{I}^*$. Therefore, for any $0 < g \leq \xi/2$ and any $(s,a)$, we know that

$$P\left( \left| \bar{\pi}^N(a|s) - \pi^*(a|s) \right| \leq \frac{2g}{\xi} \right) \geq 1 - \frac{|\mathcal{S}| + K}{g \cdot \sigma \cdot N} \cdot \left( 4 \max\{N_0, N_0'\} + 8(K + |\mathcal{S}|) \cdot \exp(-\nu^2/8) \right). \tag{124}$$

From the above inequality, for any $(s, a)$, we have that

$$\left|\mathbb{E}\left[\bar{\pi}^N(a|s)\right] - \pi^*(a|s)\right|$$

$$\leq \mathbb{E}\left[\left|\bar{\pi}^N(a|s) - \pi^*(a|s)\right|\right] \leq \frac{2}{N\xi} + \frac{2}{\xi} \cdot \int_{g=\frac{1}{N}}^{\xi/2} P\left(\left|\bar{\pi}^N(a|s) - \pi^*(a|s)\right| \geq \frac{2g}{\xi}\right) dg$$

$$\leq \frac{2}{N\xi} + \frac{2(|\mathcal{S}| + K)}{\xi \cdot \sigma \cdot N} \cdot \left(4\max\{N_0, N_0'\} + 8(K + |\mathcal{S}|) \cdot \exp(-\nu^2/8)\right) \cdot \int_{g=\frac{1}{N}}^{\xi/2} \frac{dg}{g}$$

$$= \frac{2}{N\xi} + \frac{2(|\mathcal{S}| + K)}{\xi \cdot \sigma \cdot N} \cdot \left(4\max\{N_0, N_0'\} + 8(K + |\mathcal{S}|) \cdot \exp(-\nu^2/8)\right) \cdot (\log(N) + \log(\xi/2)).$$

$$(125)$$

We finally transfer the bound (125) into the sample complexity bounds of policy $\bar{\pi}^N$. We use the state-value functions $V_r(\pi, s)$, defined for any initial state $s$ and any policy $\pi$ as follows

$$V_r(\pi, s) = \mathbb{E}\left[\sum_{t=0}^{\infty} \gamma^t \cdot r(s_t, a_t) \mid s\right], \tag{126}$$

where $(s_t, a_t)$ is generated according to the policy $\pi$ and the transition kernel $P$ with the initial state $s$. Note that the value of $V_r(\pi, s)$ for any $s \in \mathcal{S}$ can be obtained from solving Bellman's equation under policy $\pi$

$$V_r(\pi, s) = \mathbb{E}_{a \sim \pi(\cdot|s)}\left[\hat{r}(s, a) + \gamma \cdot \mathbb{E}_{s' \sim P(\cdot|s,a)}[V_r(\pi, s')]\right]. \tag{127}$$

We define a matrix $B^\pi \in \mathbb{R}^{|\mathcal{S}| \times |\mathcal{S}|}$ such that the $s$-th row $s'$-th column element is

$$B^\pi(s, s') = \delta_{s,s'} - \gamma \cdot \sum_{a \in \mathcal{A}} \pi(a|s) \cdot P(s'|s, a). \tag{128}$$

Then, the matrix $B^\pi$ represents the state transition probability matrix under the policy $\pi$. Denote by

$$\boldsymbol{V}_r(\pi) = (V_r(\pi, s))_{\forall s \in \mathcal{S}}$$

and

$$\hat{\boldsymbol{r}}(\pi) = (\sum_{a \in \mathcal{A}} \pi(a|s) \cdot \hat{r}(s, a))_{\forall s \in \mathcal{S}}.$$

We have that the state values $\boldsymbol{V}_r(\pi)$ is the solution to the linear equation

$$B^\pi \boldsymbol{V}_r(\pi) = \hat{\boldsymbol{r}}(\pi) \tag{129}$$

To bound the regret, we bound the solution to the linear equation (129) with $\pi$ being $\bar{\pi}^N$ and $\pi^*$ separately. The perturbation of the right hand of the equation (129) is

$$\Delta\hat{\boldsymbol{r}} = \hat{\boldsymbol{r}}(\bar{\pi}^N) - \hat{\boldsymbol{r}}(\pi^*).$$

Clearly, we have that

$$\|\Delta\hat{\boldsymbol{r}}\|_\infty \leq \frac{2}{N\xi} + \frac{2(|\mathcal{S}| + K)}{\xi \cdot \sigma \cdot N} \cdot \left(4\max\{N_0, N_0'\} + 8(K + |\mathcal{S}|) \cdot \exp(-\nu^2/8)\right) \cdot (\log(N) + \log(\xi/2)).$$

$$(130)$$

The perturbation of the matrix is denoted as

$$\Delta B = B^{\bar{\pi}^N} - B^{\pi^*}.$$

Clearly, it holds that

$$\|\Delta B\|_\infty \leq \frac{2\gamma}{N\xi} + \frac{2\gamma(|\mathcal{S}| + K)}{\xi \cdot \sigma \cdot N} \cdot \left(4\max\{N_0, N_0'\} + 8(K + |\mathcal{S}|) \cdot \exp(-\nu^2/8)\right) \cdot (\log(N) + \log(\xi/2)).$$

$$(131)$$

We plug the formulation of $N_0$ in (25) and $N_0'$ in (82) into the bound (130) and (131). We obtain

$$\|\Delta\hat{\boldsymbol{r}}\|_\infty \leq C_1 \cdot \frac{(|\mathcal{S}| + K)^3}{\alpha^2 \cdot \xi \cdot \sigma \cdot \min\{\sigma^2, (1-\gamma)^2 \cdot \Delta\}} \cdot \frac{\log^2(N)}{N} \tag{132}$$

where $C_1$ is a constant, $\alpha = \min_{k\in[K]}\{\alpha_k\}$, and $\Delta = \min\{\Delta_1^2, \Delta_2^2\}$ with $\Delta_1$ given in (16) and $\Delta_2$ given in (19). We also obtain

$$\|\Delta B\|_\infty \le C_1 \cdot \frac{\gamma \cdot (|\mathcal{S}| + K)^3}{\alpha^2 \cdot \xi \cdot \sigma \cdot \min\{\sigma 2, (1-\gamma)^2 \cdot \Delta\}} \cdot \frac{\log^2(N)}{N} \tag{133}$$

Therefore, as long as

$$C_1 \cdot \frac{(|\mathcal{S}| + K)^3}{\alpha^2 \cdot \xi \cdot \sigma \cdot \min\{\sigma^2, (1-\gamma)^2 \cdot \Delta\}} \cdot \frac{\log^2(N)}{N} \le 1/\|(B^{\pi^*})^{-1}\|_\infty = 1/\sigma', \tag{134}$$

following standard perturbation analysis of linear equations [28], we have that

$$\frac{\|\boldsymbol{V}_r(\bar{\pi}^N) - \boldsymbol{V}_r(\pi^*)\|_\infty}{\|\boldsymbol{V}_r(\pi^*)\|_\infty} \le C_2 \cdot \kappa(B^{\pi^*}) \cdot \left( \frac{\|\Delta B\|_\infty}{\|B^{\pi^*}\|_\infty} + \frac{\|\Delta \hat{\boldsymbol{r}}\|_\infty}{\|\hat{\boldsymbol{r}}(\pi^*)\|_\infty} \right), \tag{135}$$

where $\kappa(B^{\pi^*}) = \|B^{\pi^*}\|_\infty \cdot \|(B^{\pi^*})^{-1}\|_\infty$ denotes the conditional number of $B^{\pi^*}$, and $C_2$ is a constant. Note that we have the regret

$$\begin{aligned}
\text{Regret}_r(\bar{\pi}^N, N) = \boldsymbol{\mu}^\top(\boldsymbol{V}_r(\bar{\pi}^N) - \boldsymbol{V}_r(\pi^*)) &\le (1-\gamma)\|\boldsymbol{V}_r(\bar{\pi}^N) - \boldsymbol{V}_r(\pi^*)\|_\infty \\
&\le C_2(1-\gamma) \cdot \kappa(B^{\pi^*}) \cdot \|\boldsymbol{V}_r(\pi^*)\|_\infty \cdot \left( \frac{\|\Delta B\|_\infty}{\|B^{\pi^*}\|_\infty} + \frac{\|\Delta \hat{\boldsymbol{r}}\|_\infty}{\|\hat{\boldsymbol{r}}(\pi^*)\|_\infty} \right).
\end{aligned} \tag{136}$$

It is clear to see that

$$\|\boldsymbol{V}_r(\pi^*)\|_\infty \le \frac{1}{1-\gamma} \tag{137}$$

and

$$\|\boldsymbol{V}_r(\pi^*)\|_\infty \le \frac{\|\hat{\boldsymbol{r}}(\pi^*)\|_\infty}{1-\gamma}. \tag{138}$$

Following [37], we have the following bound.

$$\sigma' = \|(B^{\pi^*})^{-1}\|_\infty \le \frac{1}{1-\gamma}. \tag{139}$$

Also, from the definition, we have that

$$\|B^{\pi^*}\|_\infty = 1 - \gamma. \tag{140}$$

Plugging the bound (132), (133), (137), and (138), into the inequality (136), we have that

$$\text{Regret}_r(\bar{\pi}^N, N) \le C_3 \cdot \frac{(|\mathcal{S}| + K)^3}{\alpha^2 \cdot \xi \cdot \sigma(1-\gamma) \cdot \min\{\sigma^2, (1-\gamma)^2 \cdot \Delta\}} \cdot \frac{\log^2(N)}{N} \tag{141}$$

where $C_3$ is a constant. In a same way, for any $k \in [K]$, we obtain that

$$\text{Regret}_k(\bar{\pi}^N, N) \le C_3 \cdot \frac{(|\mathcal{S}| + K)^3}{\alpha^2 \cdot \xi \cdot \sigma(1-\gamma) \cdot \min\{\sigma^2, (1-\gamma)^2 \cdot \Delta\}} \cdot \frac{\log^2(N)}{N}. \tag{142}$$

To show the sample complexity bound, we denote by $\varepsilon$ a constant such that

$$\varepsilon = C_3 \cdot \frac{(|\mathcal{S}| + K)^3}{\alpha^2 \cdot \xi \cdot \sigma(1-\gamma) \cdot \min\{\sigma^2, (1-\gamma)^2 \cdot \Delta\}} \cdot \frac{\log^2(N)}{N}.$$

Therefore, we have

$$N = O\left( \frac{(|\mathcal{S}| + K)^3}{\alpha^2 \cdot \xi \cdot \sigma(1-\gamma) \cdot \min\{\sigma^2, (1-\gamma)^2 \cdot \Delta\}} \cdot \frac{\log^2(1/\varepsilon)}{\varepsilon} \right).$$

Note that in each of the $N$ rounds, we obtain a sample for each $(s, a) \in \mathcal{S} \times \mathcal{A}$. Therefore, the bound on $N$ above should be multiplied by $|\mathcal{S}| \cdot |\mathcal{A}|$ to obtain the final sample complexity bound. Our proof is thus completed.

