# OpenReview forum: "Achieving $\tilde{O}(1/\epsilon)$ Sample Complexity for Constrained Markov Decision Process"
_NeurIPS.cc/2024/Conference — NeurIPS 2024 poster_

### Official Review · Reviewer_rqcB · 2024-07-04

**Soundness:** 2
**Presentation:** 1
**Contribution:** 3
**Rating:** 6
**Confidence:** 4

**Summary:**

The paper studies the linear program formulation of constrained MDPs. The paper first characterizes the instance hardness of the underlying LP. Then, by proposing an algorithm that operates in the primal space and resolves the primal LP in an online manner, the paper derives an overall sample complexity of O(1/\epsilon) up to logarithm terms.

**Strengths:**

* The paper provides a strong bound on the constrained MDP using LP-based approaches. The obtained sample complexity is the first in the literature that achieves \tilde O(1/\epsilon).
* The derivation and presentation of theoretical results are clear and easy to follow. The proposed theoretical framework has the potential to be applied to more general online LP problems.

**Weaknesses:**

* The introduction and related work parts of the paper are not comprehensive enough (especially the related work). In the beginning of Section 1.2, the paper compares itself with [25, 29, 44, 14]. Yet, all of these works focus on policy-based methods, and I think it is not directly comparable with the proposed method in this paper. For example, the dependency of the sample complexity in this work is |S|^4, and this is usually lower for policy-based methods. Moreover, I think the paper might miss several related literature also studies occupancy-measure based approaches, for example,  "Achieving Zero Constraint Violation for Concave Utility Constrained Reinforcement Learning via Primal-Dual Approach" by Bai et al.
* The paper does not provide numerical simulations. If there is some simulation, I think it helps the reader to have a better sense of how the convergence rate of the algorithm depends on the problem size.

**Questions:**

* Can the author provide a more comprehensive literature review and problem introduction? And I think it is also important to consider the dependency of the sample complexity on the problem-related parameters (i.e., |S| and |A|) in the comparison.
* It would be better to briefly discuss the proof idea/implications of theoretical results around the theories. Currently, the ending of the paper seems too abrupt?
* If it is possible, can the author provides some preliminary numerical experiments (not mandatory)?

**Limitations:**

I did not find the place where the paper explicitly discuss the limitation (yet, the author claims "Yes" in the checklist question 2). BTW, it seems the author also forgot to provide justifications for other questions in the checklist as well.

---

> ### Author Rebuttal · Authors · 2024-08-04
>
> We would like to thank you for your insight review! Please find below our response to the weakness and your questions!
>
> $\textbf{Response to weakness 1}$: Thank you for mentioning the important references to us! We will refine our literature review part to incorporate a better comparison with existing works and approaches, including the important work you pointed out. Briefly speaking, our work presents a new algorithm. We adopt an occupancy measure representation of the optimal and obtain an LP to work with, which is similar to the previous work. However, our algorithm resolves an LP and operates in the primal space, which is fundamentally different from the previous work that adopts a primal-dual update (e.g. Stooke et al., 2020, Ding et al., 2022, Zeng et al., 2022, Bai et al., 2023, Moskovitz et al., 2023, ). There is also work developing primal-based algorithms, for example (Liu et al., 2019; Chow et al., 2018; 2019; Dalal et al., 2018, Xu et al. 2021). Our algorithm is completely different from the previous work and we obtain new results. The result is that we are able to obtain an instance-dependent $\tilde{O}(1/\epsilon)$ sample complexity the first time in the literature, which improves upon the $O(1/\epsilon^2)$ worst-case sample complexity established in the previous work. Though the constrained optimization approach and the Lyapunov approach have also been developed for CMDP problems, they do not enjoy a theoretical guarantee. In comparison to the literature, we develop a new primal-based algorithm and achieve the first instance-dependent sample complexity for CMDP problems.
>
> We will add a more comprehensive literature review. We would be appreciative if you could point out the papers that we haven't discussed and we will add discussions of those.
>
> $\textbf{Response to weakness 2}$: Thank you for the comment! We have conducted basic numerical experiments to illustrate the empirical performance of our algorithms. Please refer to the ``global response'' for more details.
>
> $\textbf{Response to question 1}$: Yeah, sure! Please find in our response to the first weakness part the refined literature review on the comparison with existing works. Also, we provide the following discussion on the dependency on other problem parameters.
>
> ``We discuss the dependency of our sample complexity bound on problem parameters other than $\epsilon$. We restrict to the MDP context without resource constraints. Denote by $\mathcal{S}$ the state set and $\mathcal{A}$ the action set. We show a sample complexity bound of $O\left( \frac{|\mathcal{S}|^4\cdot|\mathcal{A}|}{(1-\gamma)^4\cdot\Delta}\cdot\frac{\log^2(1/\epsilon)}{\epsilon}\right)$, where $\Delta$ is the constant that represents the hardness of the underlying problem instance.
> Compared to the optimal worst-case sample complexity $O\left( \frac{|\mathcal{S}|\cdot|\mathcal{A}|}{(1-\gamma)^3\cdot\epsilon^2} \right)$ that is achieved in a series of work (e.g.  Sidford et al. 2018, Wainwright 2019, Wang 2020, Agarwal et al. 2020, He et al. 2021), our bound has a worse dependency over $|\mathcal{S}|$ and $1-\gamma$. This is due to our algorithm being LP-based and the dimension of the LP ($|\mathcal{S}|$ and $1-\gamma$) will influence our final bounds. However, our bound enjoys a better dependency in terms of $\epsilon$. For the general CMDP problem, our bound will depend additionally on the conditional number of the constraint matrix in the LP formulation, which is a byproduct of the resolving LP heuristics (e.g. Vera and Banerjee 2021, Li et al. 2021). However, our $\tilde{O}(1/\epsilon)$ sample complexity bound depends polynomially on other parameters including $|\mathcal{S}|$, $|\mathcal{A}|$, $1-\gamma$, and the number of constraints.''
>
> Please note that even though our bound has some additional dependencies on other parameters, our bound can still improve upon the worst-case bound for any problem instance as long as $\epsilon$ is set to be small. To be specific, for an instance $I$, denote by $C_1(I)/\epsilon$ our bound (logarithmic term neglected) and denote by $C_2/\epsilon^2$ the worst-case bound. Then, as long as we set $\epsilon\leq C_2/C_1(I)$, our bound will be smaller. Therefore, our bound is favorable when the instance is good such that $C_1(I)$ is small, or when we are seeking for a highly accurate near-optimal solution such that $\epsilon$ is small.
>
> $\textbf{Response to question 2}$: Thanks for the comment. Due to the space limit, the discussions on the proof idea have not been added. We will for sure have the discussion. Please find below for a preliminary one.
>
> ``Our algorithm is motivated by the resolving LP heuristics in online LP/resource allocation literature (e.g. Vera and Banerjee (2021), Li and Ye (2021)). We can naturally interpret the right-hand-side constraints of the LP as the resource capacities and at each round $n$, the variables $\alpha^n$ and $\mu^n$ can be interpreted as the remaining capacities. Then, a key step in the proof is to establish that the average remaining capacities, $\alpha^n/(N-n+1)$ and $\mu^n/(N-n+1)$ behave as (sub-)martingales. As a result, we can apply concentration properties to show that the remaining capacities will diminish when we arrive at the end of the horizon, i.e., the resources have been utilized well. Moreover, since we have already identified the optimal basis in Algorithm 1 and we resolve the LP sticking to the optimal basis, we can show that when the resources are well utilized, the total reward we collect is very close to the optimal reward. In this way, we obtain a bound over the total reward collected by our policy and that of the optimal policy, which then transfers into the sample complexity bound.''
>
> $\textbf{Response to question 3}$: Thanks for the question! We are happy to add the numerical results. We have conducted basic experiments. Please refer to the ``global response'' for more details.

---

> > ### Comment · Reviewer_rqcB · 2024-08-09
> >
> > Thanks the authors for the clarification and answering my questions. I am happy to maintain my current score.

---

> > > ### Author Response · Authors · 2024-08-12
> > >
> > > Thank you so much for acknowledging our response!

---

### Official Review · Reviewer_hDFd · 2024-07-04

**Soundness:** 2
**Presentation:** 2
**Contribution:** 3
**Rating:** 5
**Confidence:** 3

**Summary:**

The paper proposes a new algorithm that solves the CMDP problem in $O(1/\epsilon)$ sample complexity, which improves the best-known $O(1/\epsilon^2)$ sample complexity in the literature. To achieve this, this paper made three contributions: (1) New characterizations of the problem instance hardness for CMDP problems. (2) A new algorithm based on LP literature instead of the traditional RL literature. (3) Extended results adopted to online LP literature.

**Strengths:**

The proposed method is new. Though it is mainly inpired by existing LP literature, to my understanding similar approaches have not been applied in RL or ML research. The LP formulation also leads to a new perspective in understanding the CMDP problem.

**Weaknesses:**

1. The presentation of this work is relatively poor. The author listed three contributions but I feel hard to understand how these contributions have contributed to this work. For example, in the Contribution 1, the author should tell (a) what is the problem instance hardness for CMDP problem, (b) what is the motivation of proposing new characterizations of the problem instance hardness for CMDP problems, (c) why the new characterization is helpful in developing the new algorithm or achieving better complexity.
2. There is no sufficient discussions on Lemma 2.1. How is it connected to the whole section Characterization of Instance Hardness?
3. The reward, cost, and transitions are all estimated during training. I cannot agree that this algorithm work with unknown transition probabilities since it requires to estimate that.
4. I doubt if the theoretical analysis is correct since it has been known that the sample complexity lower bound is $O(1/\epsilon^2)$. See [R1] and [R2].

[R1] Azar, Mohammad Gheshlaghi, et al. "Reinforcement learning with a near optimal rate of convergence." (2011).
[R2] Vaswani, Sharan, Lin Yang, and Csaba Szepesvári. "Near-optimal sample complexity bounds for constrained MDPs." Advances in Neural Information Processing Systems 35 (2022): 3110-3122.

**Questions:**

I am mainly concerned about the $O(1/\epsilon)$ complexity. Can the author clarify the difference between this work and classical RL literature?

**Limitations:**

This is a theoretical work so there is no any negative impact.

---

> ### Author Rebuttal · Authors · 2024-08-05
>
> We appreciate your insightful comments, which allow us to provide further clarifications. Please find below our response to the weakness and the questions. We hope that our response would clarify your concerns about the paper and we are happy to provide further clarifications if needed.
>
> $\textbf{Response to weakness 1}$: Thank you so much for the comment! Please allow us to further clarify. Please note that when deriving instance-dependent learning, it is important to define a measure to describe how difficult it is to separate the optimal policies from the sub-optimal ones. The importance of defining such a measure has been illustrated in other problems, such as multi-arm-bandit problems (e.g. Lai and Robbins (1985)) and reinforcement learning problems (e.g. Auer et al. (2008)). There are also other works studying how to characterize such a measure for instance-dependent learning on general sequential decision making problems, for example Wagenmaker and Foster (2023). This measure is usually defined as the gap (a positive constant) between the value of the optimal policies and the value of the best sub-optimal policies. However, since the optimal policies for CMDP problems are randomized policies, the sub-optimal policies can be arbitrarily close to the optimal ones. In our paper, we show that if we restrict the policies to the ones represented by the corner points, then such a gap can be characterized as the difference between the optimal corner points and the sub-optimal corner points. Suppose this gap is $\Delta$, then it requires $\tilde{O}(1/\Delta)$ number of samples to identify the optimal corner point.
>
> In summary, (a). ``problem instance hardness'' is a measure of the number of samples needed to separate the optimal policies from the sub-optimal ones; (b). The optimal policies for CMDP are randomized policies and thus previous measures do not apply. We need to develop a new measure that can separate the optimal randomized policies from the sub-optimal randomized policies for the CMDP problems (by restricting to the policies represented by the corner points of the LP); (c). Our new algorithm is developed based on corner point characterization. To be specific, our algorithm 1 is developed to identify one optimal corner point (one optimal basis of the LP), and our algorithm 2 resolves the LP sticking to the identified corner point to learn the optimal randomization. As we can see, identifying the optimal corner point (basis) and resolving the LP sticking to this corner point (basis) are the key elements of our algorithm, which is motivated by the instance hardness characterization via corner point.
>
> $\textbf{Response to weakness 2}$: Thank you! Lemma 2.1 is to show that for any LP, there exists an optimal basis or corner point and the LP basis or corner point can be characterized via non-zero variables and binding constraints. As discussed in our response to your previous point, this corner point representation motivates our entire approach.
>
> $\textbf{Response to weakness 3}$: Thanks for the comment. Our work assumes that the transition probabilities are unknown and need to be estimated from the data. Our algorithm is more in the sense of model-based, and this is why we need to construct estimates of the transition probabilities, as well as rewards and costs, during the execution of our algorithm.
>
> $\textbf{Response to weakness 4}$: Thank you for mentioning the two important papers! It is indeed true that if we are seeking for a $\textbf{worst-case}$ sample complexity bound, then $O(1/\epsilon^2)$ is the best we can hope for, just as illustrated in the two papers you mentioned. However, we would like to emphasize that we are deriving instance-dependent sample complexity in our paper. This is the key reason why we can break the $O(1/\epsilon^2)$ lower bound and obtain an improved $\tilde{O}(1/\epsilon)$ sample complexity. To be specific, for a problem instance $I$, we can denote by $S(I, \epsilon)$ the number of samples needed to construct an $\epsilon$-optimal policy. Then the worst-case lower bound implies that $\max_{I}S(I,\epsilon)=\Theta(1/\epsilon^2)$. However, if we do not consider the worst-case guarantee, i.e., if we do not maximize over the problem instance $I$, then we can characterize an instance-dependent constant $\Delta(I)$ (independent of $\epsilon$) such that $S(I,\epsilon)=\Delta(I)/\epsilon\cdot \text{polylog}(1/\epsilon)$. The main contributions of our paper are to (i). find a cornet-point characterization of the instance-dependent constant $\Delta(I)$, and ii). derive a policy that achieves the instance-dependent $\tilde{O}(1/\epsilon)$ sample complexity bound. Please note that our bound does not contradict with the worst-case lower bound $O(1/\epsilon^2)$. It is simply that we are seeking for an instance-dependent bound and when the problem instance is favorable such that the constant $\Delta(I)$ is smaller than $1/\epsilon$, our bound strictly improves upon the worst-case bound.
>
> $\textbf{Response to the question}$: Thank you for this question! The main difference between our work and the classical RL for CMDP literature is that we are deriving an instance-dependent sample complexity bound, while the previous literature focuses on the worst-case sample complexity bound. Indeed, the instance-dependent guarantee has been considered in the previous RL literature (but without constraints). See for example [1] and [2] for the logarithmic regret which transfers into $\tilde{O}(1/\epsilon)$ sample complexity bound. However, these works do not consider the existence of the constraints.
>
> [1]. Velegkas, Grigoris, Zhuoran Yang, and Amin Karbasi. "Reinforcement learning with logarithmic regret and policy switches." Advances in Neural Information Processing Systems 35 (2022).
>
> [2]. He, Jiafan, Dongruo Zhou, and Quanquan Gu. "Logarithmic regret for reinforcement learning with linear function approximation." International Conference on Machine Learning. PMLR, 2021.

---

> > ### Comment · Reviewer_hDFd · 2024-08-07
> >
> > Thanks for the clarification! I did misunderstand the contribution of this paper. It has been much more clear. Now I increase my score from 3 to 5.

---

> > > ### Author Response · Authors · 2024-08-08
> > >
> > > Thank you so much!

---

### Official Review · Reviewer_sc31 · 2024-07-04

**Soundness:** 3
**Presentation:** 4
**Contribution:** 3
**Rating:** 8
**Confidence:** 3

**Summary:**

This paper addresses the reinforcement learning problem for CMDPs.  The authors derived a problem-dependent sample complexity bound that is $\tilde O(1/\epsilon)$, improving upon the state-of-the-art.  They introduce a novel way to characterize the hardness of CMDP instances using the LP basis, enabling problem-dependent guarantees. The proposed algorithm involves an elimination procedure to identify an optimal basis and a resolving procedure that adapts to remaining resources, ensuring the policy remains near-optimal with fewer samples.

**Strengths:**

The paper is well written, and the intuition/ideas behind the algorithm and theoretical proofs are clearly explained, making the paper a pleasant read.  I've learned something interesting and new.

**Weaknesses:**

While this point may seem minor since the problem setting assumes a tabular formulation with finite and fully observable state and action spaces, it is important to note that the methodology becomes challenging to apply when dealing with large or infinite state spaces where function approximation is required.

**Questions:**

N/A

**Limitations:**

Suggestions:
1) A typo in 1.1 Preliminaries line 53: "stochaastic reward" -> stochastic reward
2) Define N in line 120
3) In RL, the notation $q_\pi$ is usually reserved for action-value function according to policy $\pi$.  Thus, it may be a bit disorientating for people from the RL community to see $q_\pi$ as a notation for occupancy measure. Perhaps use a different notation?  I've seen paper using $\nu$ or $d$ for defining occupancy measures.
4) On line 244, I believe "...satisfy the condition in Theorem 2.1" should be "...Lemma 2.1".

---

> ### Author Rebuttal · Authors · 2024-08-04
>
> We would like to thank you for your positive review and insightful comments! Please find below our response to your comments and questions!
>
> $\textbf{Response to weakness}$: Thank you so much for the comment! Indeed, the method developed in this paper is mainly for the tabular setting. However, the method can also be extended to handle the setting with possibly large state/action space. To be specific, we can utilize a linear function approximation and similarly write a LP with the coefficients in the linear approximation as the decision variables of the LP. We are currently working on this extension. Combining our method with more general function approximations will be a future topic for us to explore.
>
> $\textbf{Response to limitation 1}$: Thank you for pointing it out! We will correct it.
>
> $\textbf{Response to limitation 2}$: Thank you! $N$ refers to the number of episodes.
>
> $\textbf{Response to limitation 3}$: Thank you for the suggestion! Indeed, $\nu$ or $d$ is a better notation for the occupancy measure.
>
> $\textbf{Response to limitation 4}$: You are right! Thanks for the correction!

---

### Official Review · Reviewer_w3CH · 2024-07-10

**Soundness:** 2
**Presentation:** 1
**Contribution:** 2
**Rating:** 4
**Confidence:** 3

**Summary:**

The strength of this paper is that it provides strong sample complexity results for the constrained MDP, enhancing the existing analysis in the literature by developing a new algorithm. However, despite presenting a promising method, it lacks thorough comparisons with existing methods in the literature.

**Strengths:**

The strength of this paper is that it provides strong sample complexity results for the constrained MDP, enhancing the existing analysis in the literature by developing a new algorithm.

**Weaknesses:**

Although the paper presents a promising method, it does not present more through comparisons with existing methods in the literature.

**Questions:**

1) Please define the notation [K] in page 2
2) I wonder if the constraint (2) is commontly used in the literature.
Please add some discussions on the constraint (2) and if they are used in other papers.
3) The term "problem instance hardness" is frequently used in the introdcution part, but it is not familier with the most readers in my opinion. Therefore, it is necessary to clearly define what is the problem instance hardness.
4) Although the authros develop some promising algorithms, it seems that comparison with other approaches is still weak. Therefore, it would be better if some thorough discussions with existing works is added.

**Limitations:**

The authors properly addressed the limitations of the paper in the document.

---

> ### Author Rebuttal · Authors · 2024-08-04
>
> We would like to thank you for your insightful comments! Please find below our response to each of the weakness points and the questions you posted. We hope that our response would clarify your concerns about the paper.
>
> $\textbf{Response to weakness}$: Thank you for the comment! We will for sure provide a better literature review and comparison with previous methods. Briefly speaking, our work presents a new algorithm. We adopt an occupancy measure representation of the optimal and obtain an LP to work with, which is similar to the previous work. However, our algorithm resolves an LP and operates in the primal space, which is fundamentally different from the previous work that adopts a primal-dual update (e.g. Stooke et al., 2020, Ding et al., 2022, Zeng et al., 2022, Bai et al., 2023, Moskovitz et al., 2023, ). There is also work developing primal-based algorithms, for example (Liu et al., 2019; Chow et al., 2018; 2019; Dalal et al., 2018, Xu et al. 2021). Our algorithm is completely different from the previous work and we obtain new results. The result is that we are able to obtain an instance-dependent $\tilde{O}(1/\epsilon)$ sample complexity the first time in the literature, which improves upon the $O(1/\epsilon^2)$ worst-case sample complexity established in the previous work. Though the constrained optimization approach and the Lyapunov approach have also been developed for CMDP problems, they do not enjoy a theoretical guarantee. In comparison to the literature, we develop a new primal-based algorithm and achieve the first instance-dependent sample complexity for CMDP problems.
>
> We will add a more comprehensive literature review. We would be appreciative if you could point out the papers that we haven't discussed and we will add discussions of those.
>
> $\textbf{Response to question 1}$: Thanks. The definition is $[K]=\{1,\dots, K\}$.
>
> $\textbf{Response to question 2}$: Thanks for the comment! Indeed, there are other formulations of the constraints in CMDP, for example,
> \begin{equation}
> V_k(\pi, \mu_1)=\mathbb{E}\left[ \sum_{t=0}^{\infty}\gamma^t\cdot c_k(s_t, a_t)\mid \mu_1 \right] \geq \lambda_k, ~~\forall k\in[K].
> \end{equation}
> in a series of work that studies safe reinforcement learning. However, the above formulation can be transferred from our formulation in constraint $(2)$. One can set $\alpha_k=\frac{1}{1-\gamma}-\lambda_k$ for each $k\in[K]$, and it is easy to see that the two inequalities are equivalent to each other,
> \begin{equation}
> \mathbb{E}\left[ \sum_{t=0}^{\infty}\gamma^t\cdot c_k(s_t, a_t)\mid \mu_1 \right] \geq \lambda_k \Leftrightarrow \mathbb{E}\left[ \sum_{t=0}^{\infty}\gamma^t\cdot (1-c_k(s_t, a_t))\mid \mu_1 \right] \leq \alpha_k.
> \end{equation}
> Therefore, we can equivalently use the formulation in constraint $(2)$ with the cost function defined as $1-c_k$ for each $k\in[K]$.
>
> $\textbf{Response to question 3}$: Thank you! Please note that when deriving instance-dependent learning, it is important to define a measure to describe how difficult it is to separate the optimal policies from the sub-optimal ones. The importance of defining such a measure has been illustrated in other problems, such as multi-arm-bandit problems (e.g. Lai and Robbins (1985)) and reinforcement learning problems  (e.g. Auer et al. (2008)). There are also other works studying how to characterize such a measure for instance-dependent learning on general sequential decision-making problems, for example Wagenmaker and Foster (2023). This measure is usually defined as the gap (a positive constant) between the value of the optimal policies and the value of the best sub-optimal policies. However, since the optimal policies for CMDP problems are randomized policies, the sub-optimal policies can be arbitrarily close to the optimal ones. In our paper, we show that if we restrict the policies to the ones represented by the corner points, then such a gap can be characterized as the difference between the optimal corner points and the sub-optimal corner points. Suppose this gap is $\Delta$, then it requires $\tilde{O}(1/\Delta)$ number of samples to identify the optimal corner point. In summary, ``problem instance hardness'' is a measure of the number of samples needed to separate the optimal policies from the sub-optimal ones. Our corner point characterization motivates our entire approach.
>
> $\textbf{Response to question 4}$: Thank you! We will provide a better comparison with existing methods. Please refer to our response to the weakness part.

---

### Official Review · Reviewer_55Qh · 2024-07-12

**Soundness:** 4
**Presentation:** 3
**Contribution:** 4
**Rating:** 7
**Confidence:** 4

**Summary:**

This paper studies reinforcement learning problem under Constrained Markov Decision Processes (CMDPs). It formulates the problem using linear programming and designs a novel algorithm to solve it. Using the newly designed algorithm, the authors prove a sample complexity of $\tilde{O}(1/\epsilon)$, albeit at the expense of having some additional factors.

**Strengths:**

- The sample complexity analyzed in this paper breaks the barrier of $O(1/\epsilon^2)$ which is known as the lower bound for the problem that the paper studies.

- The algorithm proposed in this paper is novel. Under the linear programming framework, It designs an algorithm which only focuses on the LP basis (corner points of the feasible region). The algorithm run in $O(1/\epsilon)$ iterations and in each iteration, the order of the samples collected is independent of $\epsilon$. Unlike some traditional methods that operate in the dual space or use primal-dual techniques, this algorithm operates directly in the primal space.

**Weaknesses:**

- It has many additional dependencies such as an additional $|\mathcal{S}|^3$, $\sigma$, and etc., compared to other complexity bounds.

- Since the algorithm focuses on the corner points, it defines the separation gap $\delta_1$ between the optimal corner point and the sub-optimal corner points. They are defined to ensure the estimation errors are bounded to distinguish the optimal policy from sub-optimal policies. However, since the algorithm outputs stochastic policies, it is possible that the sub-optimal policies are very close to the optimal policy. Similarly, $\delta_2$ is the minimum gap in the dual values when some constraints are excluded. If the constraints do not change the value of the dual problem much, $\delta_2$ will also be small. Therefore, an additional $\delta = \min \{\delta_1^2, \delta_2^2 \}$ term might not be worth substituting for $\epsilon$.

- In addition, from the definitions of $\xi$ and $\sigma$, it is very likely they will be small terms. For example, for rarely visited states in $q^*$ and small eigenvalues for $A^*$.

- Due to the above concerns, it would be better to conduct experiments showing that the samples needed under such a framework are indeed fewer than those needed for other algorithms achieving $O(1/\epsilon^2)$. However, there are no numerical experiments done in this work. Thus, it is hard to tell whether the newly proposed algorithm is more efficient or not in reality.

**Questions:**

Could you theoretically provide some cases to illustrate when those additional dependencies do not make the bound worse? Besides, it will better if the cases provided are not edge cases.

**Limitations:**

There are no potential negative societal impact concerns for this theoretical work.

---

> ### Author Rebuttal · Authors · 2024-08-04
>
> We would like to thank you for your positive review and insightful comments! Please find below our response to the weakness and your question. We hope that our response would clarify your concerns regarding our work.
>
> $\textbf{Response to weakness 1}$: Thank you for the comment! You are right that our bound has additional dependencies on the model parameters, however, this seems to be common for instance-dependent learning and has shown up in other works. It is easy to understand that when we obtain a better dependency on $\epsilon$, we would suffer from a worse dependency on other parameters. However, it is important to note that the additional dependency on the model parameters is fixed and is independent of $\epsilon$. Therefore, when we are seeking a highly accurate near-optimal policy and set $\epsilon$ to be small, our bound will be a better one. Please further refer to our response to your question for a more detailed explanation.
>
> $\textbf{Response to weakness 2}$: Thanks and you are right that for some instances, the parameter $\delta$ can be small, making it difficult to separate the optimal policy from the sub-optimal ones and leading to a large bound. However, please note that no matter how small $\delta$ can be, the parameter $\delta$ is independent of $\epsilon$. Therefore, even if $\delta$ is very small, as long as we are seeking a solution with high accuracy, i.e., the error term $\epsilon$ is also very small, it could still be desirable to use $\delta$ to substitute $\epsilon$.
>
> $\textbf{Response to weakness 3}$: Thanks for the comments! You are right that for some bad instances, the dependencies on other problem parameters can be large. However, these parameters are always independent of $\epsilon$. Therefore, when we want to find a near-optimal policy with a small $\epsilon$, the $\tilde{O}(1/\epsilon)$ bound is desirable even if the dependencies on other parameters are large. Please refer to our response to your question below for more detailed explanations.
>
> $\textbf{Response to weakness 4}$: You are right that it is better to conduct some numerical experiments to support our results. Due to the time limit, we conduct basic experiments. Please refer to the ``global response'' for more details on our numerical experiments.
>
> $\textbf{Response to the question}$: Thank you so much for the question which allows us to provide further clarifications! Please note that though our bound has some additional dependencies on the problem parameters, they are independent of the accuracy level $\epsilon$. To be specific, for a problem instance $I$, denote by $C_1(I)$ the constant term in our bound. Then, our sample complexity bound is $C_1(I)\cdot\log^2(1/\epsilon)/\epsilon$ on the instance $I$. The worst-case sample complexity bound established in the previous literature is $C_2/\epsilon^2$. Please note that $\epsilon$ is the accuracy level that we can decide. Therefore, for any problem instance $I$, as long as we set $\epsilon$ small enough such that $\epsilon\leq O(C_2/C_1(I))$, our instance-dependent bound $C_1(I)\cdot\log^2(1/\epsilon)/\epsilon$ will be better than the worst-case bound $C_2/\epsilon^2$. That being said, for any problem instance, even if the problem instance is not that favorable such that the constant term in our bound is large, our bound can always be better than the worst-case $O(1/\epsilon^2)$ as long as we set $\epsilon$ small enough, i.e., we are seeking for a policy with low error.

---

> > ### Comment · Reviewer_55Qh · 2024-08-14
> > **Thank you.**
> >
> > Thank you for the clarification and my questions are mostly addressed. I will keep my score the same.

---

> > > ### Author Response · Authors · 2024-08-14
> > >
> > > Thank you for acknowledging our response!

---

### Author Rebuttal · Authors · 2024-08-06

We implement our algorithm to study the numerical performance. We consider a CMDP problem with the state space $|\mathcal{S}|=10$ and the action space $|\mathcal{A}|=10$. We set the discount factor $\gamma=0.7$. We then randomly generate the probability transition kernel $P$. To be specific, for each state $s\in\mathcal{S}$, action $a\in\mathcal{A}$, and the future state $s'\in\mathcal{S}$, we uniformly generate a randomly variable $p_{s,a,s'}$. Then, the transition probability is defined as $P(s'|s,a)=\frac{p_{s,a,s'}}{\sum_{s''\in\mathcal{S}}p_{s,a,s''}}$. For each state-action pair $(s,a)\in\mathcal{S}\times\mathcal{A}$, the expected reward $\hat{r}(s,a)$ is uniformly generated from the interval $[1,2]$ (with the reward for the first action set to be $0$). The actual reward $r(s,a)=\hat{r}(s,a)+\eta$,  where $\eta$ is uniformly distributed among $[-0.5, 0.5]$. There are $K=5$ constraints and for each constraint $k\in[K]$ and each state-action pair $(s,a)\in\mathcal{S}\times\mathcal{A}$, we define the expected cost $\hat{c}_k(s,a)$ to be uniformly generated from $[1,2]$. The actual cost $c_k(s,a)=\hat{c}_k(s,a)+\eta'$, where $\eta'$ is uniformly distributed among $[-0.5, 0.5]$.

For each total iterations $N$, We apply our algorithm and obtain the output $q^1, \dots, q^N$. We compare $\bar{q}^N$ with the optimal occupancy measure and we define the error term as $\text{Err}(N)=\|\bar{q}^N-q^*\|_{1}/\|q^*\|_1$. We study how the error term $\text{Err}(N)$ scales with $N$. The results are displayed in the attached PDF. As we can see, the error term drops to $0.02$ within $N=5000$ iterations and it keeps improving as we have more and more iterations. The computation time of our algorithm is also fast in that in each iteration, we only need to solve a set of linear equations. These evidences demonstrate the numerical efficiency of our algorithm and we will conduct more involved experiments in the future work.

---

### Decision · Program_Chairs · 2024-09-25

**Decision:**

Accept (poster)

**Comment:**

This paper derives optimal problem-dependent guarantees for CMDPs. In particular, the paper proposes a primal only algorithm and prove that the resulting algorithm has an instance-dependent $O(1/\epsilon)$ sample complexity bound.

The reviewers agree that the paper is well-written, and its contributions merit acceptance. After carefully reading the paper and the corresponding discussion, I tend to agree. Please incorporate the reviewers' feedback. In particular, addressing the following concerns will help strengthen the paper:
- Include the experiment done as part of the author response. An empirical comparison with an existing approach (e.g. primal-dual policy gradient) will be helpful.
- Explain whether the proposed algorithm (without any modification) retains the $O(1/\epsilon^2)$ problem-independent sample-complexity
- Better situate the resulting algorithm and compare to the existing literature (response to Rev. w3CH)
- Add the discussion about the minimax lower-bound (response to Rev. hDFd)
- Compare to the previous bounds in terms of the dependence in $S$, $A$ (and not just $\epsilon$)
- Add a proof sketch. Including some more details about the online LP literature and better connecting it to the CMDP problem will enhance the paper's readability for the audience working on constrained RL.
- Clearly explain the dependence on the constants in Theorem 5.2. What is tight and what can be relaxed? Conjecturing a lower bound and having a discussion section will be helpful.